# DIVERSE AND CONSISTENT MULTI-VIEW NETWORKS FOR SEMI-SUPERVISED REGRESSION

## ABSTRACT

Label collection is costly in many applications, which poses the need for label-efficient learning. In this work, we present Diverse and Consistent Multi-view Networks (DiCoM) – a novel semi-supervised regression technique based on a multi-view learning framework. DiCoM combines diversity with consistency – two seemingly opposing yet complementary principles of multi-view learning - based on underlying probabilistic graphical assumptions. Given multiple deep views of the same input, DiCoM encourages a negative correlation among the views' predictions on labeled data, while simultaneously enforces their agreement on unlabeled data. DiCoM can utilize either multi-network or multi-branch architectures to make a trade-off between computational cost and modeling performance. Under realistic evaluation setups, DiCoM outperforms competing methods on tabular and image data. Our ablation studies confirm the importance of having both consistency and diversity.

## 1 INTRODUCTION

Deep neural networks have achieved tremendous success across several domains, ranging from computer vision, natural language processing, to audio analysis (LeCun et al., 2015). However, to train neural networks that perform well typically requires a large amount of labeled data. In many cases, this requirement for a large labeled dataset presents a challenge, because the annotation process can be labour-intensive and thus expensive, especially when specialized expertise is required. To address this challenge, semi-supervised learning methods (Van Engelen & Hoos, 2020) that can achieve similarly high performance with less labeled data by using unlabeled data have been developed.

We focus on semi-supervised learning in the regression setting. There are several approaches for semi-supervised regression, including graph-based methods (Zhur & Ghahramanirh, 2002), co-training (Blum & Mitchell, 1998) and entropy minimization (Jean et al., 2018). Consistency-based approaches that have been popular in the classification setting, such as Mean Teacher (Tarvainen & Valpola, 2017) and Virtual Adversarial Training (Miyato et al., 2018), which reinforce the output consistency of the network under input perturbations, have also been adapted to the regression setting (Jean et al., 2018). However, enforcing consistency alone may not be sufficient for good performance, and may lead to model collapsing (Qiao et al., 2018) or confirmation bias issues (Ke et al., 2019).

To address these issues, we draw inspiration from ensemble learning with neural networks for regression. A necessary and sufficient condition for an ensemble of learners to be more accurate than any of its individual members is if the base learners are accurate and diverse (Dietterich, 2000). Therefore, the key component that can make or break an ensemble is the diversity (or disagreement) among its individual regressors. If this diversity is insufficient, the ensembling may not result in better performance. On the other hand, overemphasizing diversity can degrade the learnability of the ensemble members. So far, the most successful mechanism to leverage ensemble diversity in regression is Negative Correlation Learning (Liu & Yao, 1999; Zhang et al., 2019).

In this work, we propose Diverse and Consistent Multi-view Networks for Semi-supervised Regression (DiCoM) that elegantly unifies consistency and diversity in a multi-view learning framework. Based on probabilistic graphical assumptions, we derive a loss function that integrates both consistency and diversity components – diversity is encouraged on labeled data, while consistency is enforced on unlabeled data. Furthermore, we develop two variants of DiCoM: the first uses multiple

networks to achieve better performance, while the second employs a single network with multiple branches to help with scalability. We compare DiCoM against state-of-the-art methods on eight tabular datasets and a crowd-counting dataset, where we show that DiCoM outperforms existing methods. We further perform ablation studies to analyze the importance of diversity and consistency, and the effect of varying the number of views in the model.

While other works have leveraged related ideas of complementary and consensus in multi-view classification (Xu et al., 2013); or explored commonality and individuality in multi-modal curriculum learning (Gong, 2017), these methods were developed for classification or clustering tasks, and cannot be easily modified to suit semi-supervised regression.

In summary, the major contributions of this work are as follows:

- We derive a novel objective function from a probabilistic graphical perspective. Our objective function unifies multi-view diversity and consistency and provides theoretical insights into the relationship between diversity-consistency trade-off and the number of views.

- We show the high flexibility of DiCoM, which can be adaptively scaled up to larger number of views while maintaining competitive performance.

- We demonstrate the performance of DiCoM on both tabular and visual types of input data, where it outperforms competing methods. Our ablation studies validate the importance of having both diversity and consistency.

## 2 RELATED WORK

**Semi-supervised Regression:** Semi-supervised learning is a data-efficient learning paradigm that offers the ability to learn from unlabeled data. In recent years, much work has focused on semi-supervised classification, and there have been far fewer studies on semi-supervised regression. For regression tasks, graph-based methods are among the first to be developed. One example is Label Propagation (LP) (Zhur & Ghahramanirh, 2002) which defines a graph of training data and propagates ground-truth labels through high density regions of the graph. Kernel methods have also been proposed, such as Semi-supervised Deep Kernel Learning (SSDKL) (Jean et al., 2018). This method minimizes the predictive variance in a posterior regularization framework to learn a more generalizable feature embedding on unlabeled data. Co-training regressors (COREG) (Zhou & Li, 2005) employs k-Nearest neighbor regressors, each of which generate pseudo-labels for the other during training; this helps to maximize their agreement on unlabeled data.

Apart from the aforementioned approaches, *consistency-based methods* are also gaining traction. Mean Teacher (MT) (Tarvainen & Valpola, 2017) enforces posterior consistency between two neural networks, a student and a teacher, the latter being an exponential moving average of the former in the parameter space. An orthogonal approach is to enforce consistency on adversarially augmented input, as implemented in Virtual Adversarial Training (VAT) (Miyato et al., 2018). These methods were originally developed for classification, and were subsequently adapted to regression tasks (Jean et al., 2018). However, both MT and VAT maintain only a single trainable network, which may lead to problems such as confirmation bias (Ke et al., 2019) and overly-sensitive hyperparameters. In this paper, we show that consistency-based methods can be further improved with ensemble diversity.

**Ensemble Diversity:** Ensembles of neural networks have been extensively studied and widely used in many applications. Their effectiveness largely depends on the level of diversity (or disagreement) among members of the ensemble. It is well-understood that a good ensemble must manage the trade-off between the accuracy of the individual learners and the diversity among them (Brown et al., 2005; Tang et al., 2006). For regression tasks, a commonly-used ensemble technique is Negative Correlation Learning (NCL) (Liu & Yao, 1999; Liu et al., 2000), which formulates a diversity-promoting loss using an *ambiguity decomposition* of the squared ensemble loss (Krogh et al., 1995). In this formulation, a correlation penalty term (also refered to as an ambiguity term) measures how much each member's prediction deviates from the ensemble output. When this penalty term is maximized, the errors of individual learners become negatively correlated. It was theoretically proven (Brown et al., 2005) that the strategy employed by NCL is equivalent to leveraging a *bias-variance-covariance trade-off* (Ueda & Nakano, 1996) of the ensemble error.

Recently, NCL has been extended to semi-supervised learning (Chen et al., 2018), where the correlation penalty term is extended to the unlabeled data. However, this method was demonstrated only on tabular data. Another variant of NCL is Deep Negative Correlation Learning (DNCL) (Shi et al., 2018; Zhang et al., 2019), which is designed for visual regression tasks in a purely supervised learning setting.

**Multi-view Learning:** A dataset is considered as having multiple *views* when its data samples are represented by more than one feature set, each of which is sufficient for the learning task. Although each view is supposed to be sufficient for learning the task, a model trained on only one single view often faces the risk of overfitting, especially when labeled data is limited (Xu et al., 2013). To address this problem, multi-view learning assigns a modeling function to each view and jointly optimize these functions to improve overall generalization performance (Zhao et al., 2017). By analyzing the development of various techniques, Xu *et al*. (Xu et al., 2013) summarized two significant principles that underpin multi-view learning: *consensus* and *complementary*. The consensus principle states that a multi-view technique must aim at maximizing the agreement on different views. This is similar to how consistency-based semi-supervised learning methods works: for instance, MT enforces agreement with its past self. The complementary principle states that in order to make improvement, each view must contain some information that the other views do not carry. In other words, the views should be sufficiently diverse. This is related to diversity regularization in ensemble learning, where individual learners are encouraged to give diverse predictions. Thus, multi-view learning offers a unifying perspective of both consistency and diversity.

## 3 PROPOSED METHOD

We start by describing how multiple deep views can be generated from input data. Then, we propose our multi-view learning framework for regression, in which multiple deep views can be simultaneously optimized via backpropagation. We then discuss the graphical models that govern the probabilistic dependencies among the ground-truth label and the deep views. Finally, we derive the DiCoM loss function using these graphical models and provide a few insights.

**View creation:** Consider a regression task where the goal is to estimate a label $y \in \mathbb{R}$ from an input $x$. To create multiple views, our first approach is to use $M$ neural networks $F_1, F_2, \ldots, F_M$, each parameterized by $\theta_1, \theta_2, \ldots, \theta_M$, respectively. By applying different data augmentations $\eta_1, \eta_2, \ldots, \eta_M$ on the original $x$, we generate $M$ different augmented inputs $x^m = \eta_m(x) \ \forall \ m = 1, \ldots, M$. With each augmented input, the corresponding neural network produces a regression output $f_m(x) = F_m(x^m, \theta_m) \ \forall \ m = 1, \ldots, M$. Due to the different augmentations and network parameters, each output $f_m$ can be treated as one *deep view* of the original input $x$. We call this multi-network setup DiCoM-N, where the N stands for 'network' (see Fig. 1 (a)).

The second approach is to utilize a single network with a shared backbone $B$ and multiple branches $F_1, F_2, \ldots, F_M$. We use $\theta_B$ to denote the learnable parameters from the backbone and $\theta_1, \theta_2, \ldots, \theta_M$ to denote the parameters of the branches. The hidden features generated by the backbone serve as input to the branches. While the backbone still applies a random augmentation to the input $x$, each branch $F_m$ applies its own random augmentation $\eta_m$ as well. Thus, regression outputs $f_1, f_2, \ldots, f_M$ from the branches can be considered as deep views of the original input. We name this setup DiCoM-B, where the B is short for 'branch' (see Fig. 1 (b)). Multi-branch technique was widely adopted for supervised classification (Xie et al., 2017). In this work, it allows us to harness the power of multi-view learning with a relatively lower number of trainable parameters.

**Multi-view learning framework for regression:** Regardless of how they are generated, the deep views are used together with the true label $y$ to compute a semi-supervised loss function $\mathfrak{L}_{\text{DiCoM}}$. During training phase, $\mathfrak{L}_{\text{DiCoM}}$ is back-propagated simultaneously through the deep views to optimize network parameters $\theta_1, \theta_2, \ldots, \theta_M$ (including $\theta_B$ in the case of DiCoM-B). During inference, all augmentations are removed so that the forward pass is applied on the raw input $x$. The final prediction is computed as the average of all deep views: $\mu(x) = \sum_{i=1}^{M} \frac{1}{M} f_m(x)$. The general DiCoM framework is illustrated in Fig. 1. In the next step, we derive $\mathfrak{L}_{\text{DiCoM}}$ based on a probabilistic graphical assumption.

**Probabilistic graphical models:** Since the augmented inputs are generated from the same sample, the deep views should be close to each other. Motivated by previous work in kernel learning (Yu

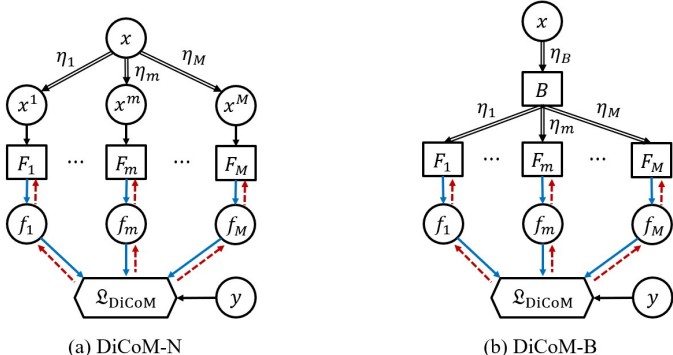

Figure 1: The DiCoM framework with two variants: (a) DiCoM-N (multi-network) and (b) DiCoM-B (multi-branch). Solid black arrows denote input/output; double black arrows denote augmentations; solid blue and dotted red arrows represent forward and backward passes, respectively.

et al., 2011) and linear regression (Nguyen et al., 2019), we consider $f_1, f_2, \ldots, f_M$ as random variables and introduce a *consensus function* $f_c$ as a latent variable that connects to each of the deep views. This function enforces the mutual agreement among the views. We assume that the difference between the consensus function and each view follows a zero-mean Gaussian distribution

$$f_c - f_m \propto \mathcal{N}\left(0, \sigma_m^2\right) \quad \forall m = 1, \ldots, M. \tag{1}$$

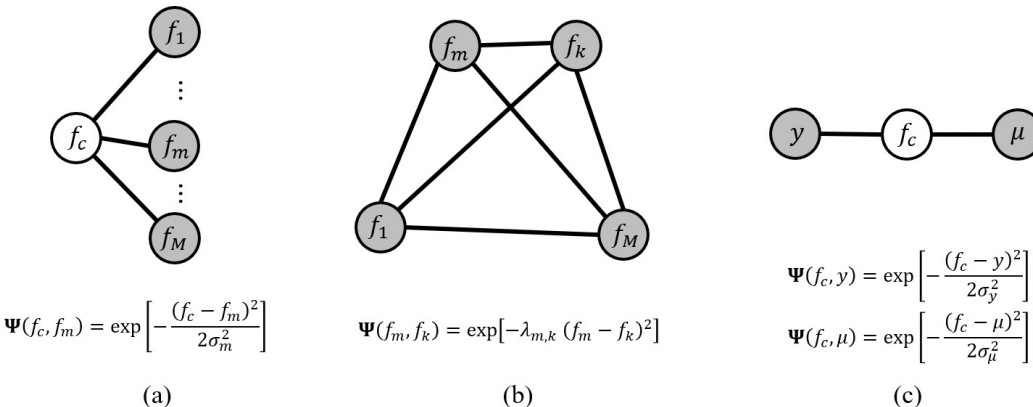

Figure 2: Undirected probabilistic graphical models of DiCoM: (a) for an unlabeled sample, (b) after marginalization of the views and (c) for a labeled sample.

This probabilistic relation is known as the *consensus potential* (Yu et al., 2011). Considering the whole graph, this potential implies that all views are random Gaussian variables with a shared mean $f_c$ and variance $\sigma_m^2$. As a result, the views stay consistent w.r.t. each other by taking values not too far away from the shared consensus. This graphical model, shown in Fig. 3 (a), is assumed for each unlabeled sample. The joint density associated with the graph is given by

$$p\left(f_c, f_1, \ldots, f_M\right) = \frac{1}{\mathcal{Z}} \prod_{m=1}^{M} \Psi\left(f_c, f_m\right) \tag{2}$$

where $\mathcal{Z}$ is a normalizing constant and $\Psi\left(f_c, f_m\right) = \exp\left[-\frac{(f_c - f_m)^2}{2\sigma_m^2}\right]$ is the potential function of the edge connecting $f_c$ and $f_m$. From this model, we derive two important results. On a side note,

our derivation generalizes to vector-valued labels $y$, but here we assume scalar labels for ease of exposition. The proofs of our results are provided in Appendix A.

**(I) Marginalization of the views:** By integrating the latent consensus function $f_c$ out of the joint density, the marginal distribution of the views is

$$p\left(f_1, \ldots, f_M\right) \propto \exp\left[\sum_{m=1}^{M} \sum_{k>m} -\lambda_{m,k} \left(f_m - f_k\right)^2\right] \tag{3}$$

where $\lambda_{m,k} = \left[2\sigma_m^2 \sigma_k^2 \left(\sum_m \frac{1}{\sigma_m^2}\right)\right]^{-1}$. This result implies that the marginal likelihood can be factorized as a product of $\binom{M}{2}$ terms. Each term is an isotropic Gaussian distribution on the difference between a pair of views $(f_m, f_k)$, with zero mean and variance $(2\lambda_{m,k})^{-1}$. The equivalent graphical model is shown in Fig. 3(b).

**(II) Conditional of the consensus function:** By applying Bayes' theorem, the conditional distribution of the consensus function $f_c$ given all the views is a Gaussian

$$f_c|f_1, \ldots, f_M \sim \mathcal{N}\left(\tilde{\mu}, \sigma_\mu^2\right). \tag{4}$$

where $\sigma_\mu^2 = \left(\sum_m \frac{1}{\sigma_m^2}\right)^{-1}$ and $\tilde{\mu} = \sigma_\mu^2 \sum_m \frac{f_m}{\sigma_m^2}$. This result highlights that the conditional distribution of $f_c$ depends only on the weighted average $\tilde{\mu}$, and the values of individual views are not required. Furthermore, $\tilde{\mu}$ can be treated as a view itself, with a variance that is smaller than any of the variances of the views.

**Derivation of DiCoM loss function:** For simplicity, we assume equal variance for different deep views, i.e., $\sigma_m^2 = \sigma_v^2 \quad \forall m$. For an unlabeled sample $(x_n)$, we directly apply the first result **(I)** to obtain the following negative log likelihood function

$$L_{\text{unl}} = \sum_{m=1}^{M} \sum_{k>m} \frac{1}{2M\sigma_v^2} \left[f_m(x_n) - f_k(x_n)\right]^2 \tag{5}$$

For a labeled sample $(x_n, y_n)$, since the ground-truth is given, we assume a graphical model that involves the final DiCoM prediction, i.e., the averaged output $\mu$. This graph is shown in Fig. 3(c). Since we assume a shared variance $\sigma_v^2$, the weighted output now reduces to an equal-weight average, following from result **(II)**

$$\tilde{\mu}(x_n) = \sum_{m=1}^{M} \frac{f_m(x_n)}{M} = \mu(x_n) \quad \sigma_\mu^2 = \frac{\sigma_v^2}{M} \tag{6}$$

Subsequently, we apply result **(I)** on this graph to get the negative log likelihood as follows

$$L_{\text{lab}} = \frac{M}{2M\sigma_y^2 + 2\sigma_v^2} \left[y_n - \mu(x_n)\right]^2 \tag{7}$$

$$= \frac{1}{2M\sigma_y^2 + 2\sigma_v^2} \sum_{m=1}^{M} \left\{\left[f_m(x_n) - y_n\right]^2 - \left[f_m(x_n) - \mu(x_n)\right]^2\right\} \tag{8}$$

$$\approx \frac{1}{2\sigma_v^2} \sum_{m=1}^{M} \left\{\left[f_m(x_n) - y_n\right]^2 - \left[f_m(x_n) - \mu(x_n)\right]^2\right\} \tag{9}$$

where in equation (8), we have applied the *ambiguity decomposition* (Krogh et al., 1995) and in equation (9), we have assumed that the label is accurate, i.e., $\sigma_y^2 \ll \sigma_v^2$.

Given a training batch of labeled samples $\{(x_n, y_n)\}_{n=1}^{L}$ and unlabeled samples $\{(x_n)\}_{n=1}^{U}$, assuming that the samples are independently generated, we can add the log-likelihood functions across all training samples. This can be done via simply adding up two equations (5) and (9)

$$\mathfrak{L}_{\text{DiCoM}} = \frac{1}{L} \sum_{n=1}^{L} \sum_{m=1}^{M} \left\{\left[f_m(x_n) - y_n\right]^2 - \kappa_{\text{div}} \left[f_m(x_n) - \mu(x_n)\right]^2\right\}$$

$$+ \frac{1}{U} \sum_{n=1}^{U} \sum_{m=1}^{M} \sum_{k>m} \kappa_{\text{csc}} \left[f_m(x_n) - f_k(x_n)\right]^2 \tag{10}$$

where we introduce two hyperparameters $\kappa_{\text{div}}$ and $\kappa_{\text{csc}}$ to absorb other constants and to enable a trade-off between diversity and consistency components of the loss.

The DiCoM loss encourages *diversity on labeled data*, while enforcing *consistency on unlabeled data*. These two seemingly opposing components can both be derived from the same underlying graphical assumptions. Furthermore, they should not be weighted equally. In fact, we have shown that it depends on the number of views: when M increases, diversity grows in $O(M)$, while consistency grows in $O(M^2)$. It is worth noting that our method is fundamentally different from other extensions of NCL such as Semi-supervised NCL (Jean et al., 2018), which enforces diversity on both labeled and unlabeled data. Last but not least, since both diversity and consistency are incorporated in the DiCoM objective function, the method is highly adaptable to different implementations such as multi-network or multi-branch, as long as the views are provided.

## 4  EXPERIMENTS

In this section, we study the proposed method in different settings, including regression tasks on eight tabular datasets and a crowd counting task on image data. We provide additional experiment results in Appendix C and an additional experiment on toy data in Appendix D.

### 4.1  EXPERIMENTS ON UCI TABULAR DATA

**Datasets:** We evaluate DiCoM on eight datasets from the UCI repository (Dua & Graff, 2017)[1]: skillcraft, parkinsons, elevators, protein, blog, ctslice, buzz, and electric. These datasets are collected from real-world regression scenarios, with varying sample sizes and input dimensions. For each dataset, we keep 1000 labeled samples as a hold-out test set; further retain $N = 300$ samples for the labeled training set, and keep the rest as the unlabeled training set. We follow the realistic evaluation setup in (Oliver et al., 2018) and use a 90%-10% train-validation split, i.e., 270 samples are used for training, leaving only 30 for validation.

**Experiment Setup:** We implement both variants of DiCoM. Our DiCoM-N networks adopt the same architecture as (Jean et al., 2018; Wilson et al., 2016), which is a fully-connected multilayer perceptron with four hidden layers, containing 100, 50, 50 and 2 hidden nodes, respectively. Our DiCoM-B also utilizes this architecture, but branch out after the third hidden layer, i.e., the backbone contains hidden layers of 100, 50 and 50 nodes, while the branches each contains one hidden layers of 2 nodes. This model is trained end-to-end, the backbone is trained together with the branches. DiCoM hyperparameters $(\kappa_{\text{div}}, \kappa_{\text{csc}})$ are chosen from a grid of values based on validation errors. Across 10 random seeds, we report the root-mean-squared errors (RMSE) statistics on the test set. For simplicity, we append '-$M$' to the end of our method name to denote the number of views, e.g., DiCoM-B-4 represents the multi-branch DiCoM network with 4 branches.

**Data Augmentation:** we apply zero-mean Gaussian noise which is commonly used for tabular data. For the DiCoM-B model, Gaussian noise is applied on the input and on the features at the beginning of each branch, right after branching out. Since the independent Gaussian noise is added during the forward pass, it does not affect the gradient values during backpropagation. The exact amount of Gaussian noise can be found in Appendix B.

We compare DiCoM against six semi-supervised regression methods: SSDKL (Jean et al., 2018), COREG (Zhou & Li, 2005), LP (Zhur & Ghahramanirh, 2002), VAE (Jean et al., 2018), MT (Tarvainen & Valpola, 2017) and VAT (Miyato et al., 2018). These methods span a wide range of approaches such as consistency regularization (MT, VAT), entropy minimization (SSDKL), multi-view learning (COREG), graph-based (LP), or generative modeling (VAE). The detailed information about the datasets and experiment setup can be found in Appendix B.

**Results:** Fig. 3 shows the experiment results. We observe significant improvements compared to the state-of-the-art semi-supervised regression methods. The largest performance gains are achieved on parkinsons and ctslice, where DiCoM-N-2 improves upon the best competing method by 32.4% and 42.5%, respectively. DiCoM-N-2 also outperforms DiCoM-B-2 on all datasets, except for protein. This is expected since DiCoM-N-2 has almost twice as many learnable parameters as DiCoM-B-2. For elevators, the most frequently selected hyperparameters across 10 ran-

---

[1] https://archive.ics.uci.edu/ml/index.php

dom seeds are $(\kappa_{\mathrm{div}}, \kappa_{\mathrm{csc}}) = (1, 0.01)$, while for ctslice, the most frequently selected values are $(\kappa_{\mathrm{div}}, \kappa_{\mathrm{csc}}) = (0.1, 1)$. This shows that different datasets require different trade-offs between consistency and diversity. We also notice that LP (a graph-based method) and COREG (a nearest-neighbors-based method) performs relatively well on blog, ctslice and buzz. Meanwhile, MT and VAT, which are based on consistency regularization (without diversity regularization), did not perform well on these regression datasets, even though they have been shown to be effective on classification tasks.

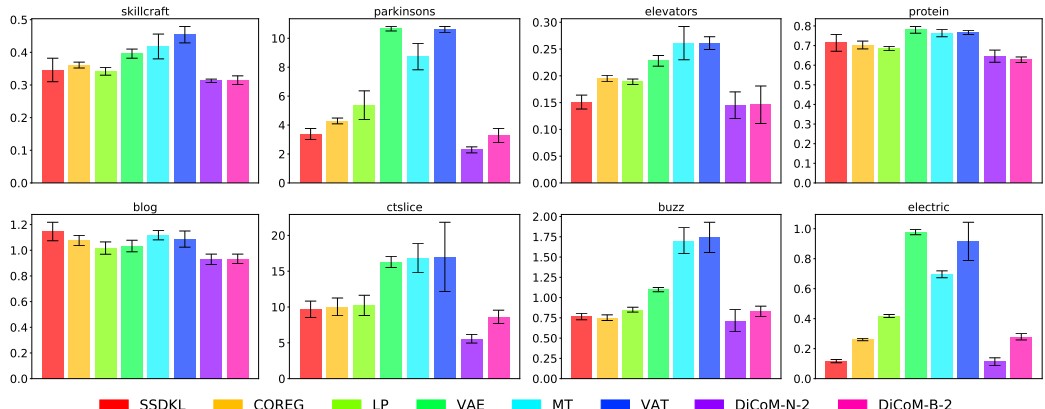

Figure 3: Test RMSE on UCI datasets: each subplot shows the results for one dataset.

**Ablation study on the components:** We analyse the effect of different components of the DiCoM-N-2 model by individually removing them from the model. For the first model, Ablation-1, we remove data augmentation. In Ablation-2, we remove the diversity loss on the unlabeled data. Next, the consistency loss on labeled data is set to zero for Ablation-3 model. Lastly, we apply diversity loss to both labeled and unlabeled training data for model Ablation-4. Using the results of DiCoM-N-2 as the baseline, we also report the percentage reduction in test RMSE (% Redc.) for the other methods as follows:

$$\text{Percentage Reduction} = \frac{(\text{baseline score} - \text{new score}) \times 100}{\text{baseline score}} \quad (11)$$

Table 1 reports the ablation results. The average percentage reduction scores tell us the importance of each component. Augmentation has a small impact on the performance of DiCoM-N-2, while diversity and consistency regularization are both important. DiCoM-N-2 outperforms Ablation-4 in all cases, which suggests that a mere reliance on diversity is insufficient.

Table 1: Test RMSE from Ablation Study on UCI Datasets.

|  | DiCoM-N-2 | Ablation-1 | | Ablation-2 | | Ablation-3 | | Ablation-4 | |
|---|---|---|---|---|---|---|---|---|---|
| **Aug.** | ✓ | | | ✓ | | ✓ | | ✓ | |
| **Lab. Div.** | ✓ | ✓ | | | | ✓ | | ✓ | |
| **Unlab. Csc.** | ✓ | ✓ | | ✓ | | | | | |
| **Unlab. Div.** | | | | | | | | ✓ | |
|  | RMSE | RMSE | % Redc. | RMSE | % Redc. | RMSE | % Redc. | RMSE | % Redc. |
| **skillcraft** | **0.313 ± 0.005** | 0.330 ± 0.017 | -5.559 | 0.327 ± 0.008 | -4.610 | 0.333 ± 0.012 | -6.446 | 0.330 ± 0.011 | -5.326 |
| **parkinsons** | **2.285 ± 0.208** | 2.437 ± 0.280 | -6.666 | 2.390 ± 0.278 | -4.593 | 2.563 ± 0.299 | -12.159 | 2.438 ± 0.286 | -6.692 |
| **elevators** | 0.145 ± 0.025 | **0.142 ± 0.031** | 2.082 | 0.149 ± 0.022 | -2.328 | 0.155 ± 0.027 | -6.225 | 0.157 ± 0.028 | -7.936 |
| **protein** | **0.646 ± 0.031** | 0.679 ± 0.029 | -5.153 | 0.654 ± 0.025 | -1.313 | 0.669 ± 0.034 | -3.546 | 0.661 ± 0.030 | -2.327 |
| **blog** | **0.930 ± 0.040** | 1.014 ± 0.039 | -8.998 | 1.021 ± 0.051 | -9.752 | 1.027 ± 0.036 | -10.455 | 1.007 ± 0.045 | -8.261 |
| **ctslice** | **5.575 ± 0.606** | 6.976 ± 0.984 | -25.120 | 7.154 ± 1.176 | -28.306 | 7.461 ± 1.014 | -33.813 | 7.398 ± 0.662 | -32.699 |
| **buzz** | **0.715 ± 0.136** | 0.757 ± 0.064 | -5.825 | 0.938 ± 0.489 | -31.184 | 0.830 ± 0.131 | -16.045 | 1.038 ± 0.550 | -45.085 |
| **electric** | 0.114 ± 0.025 | **0.097 ± 0.026** | 14.658 | 0.153 ± 0.124 | -33.833 | 0.107 ± 0.011 | 6.327 | 0.150 ± 0.117 | -31.543 |
| **Average** | | | -5.073 | | -14.490 | | -10.295 | | -17.484 |

**Varying the number of views:** We further evaluate the impact of the number of views $M$. Fig. 4 shows the performance of three DiCoM-N models with increasing number of views $M \in \{2, 4, 8\}$. The results show that a larger value of $M$ leads to an improvement in the performance. When $M$ increases from 4 to 8, the average reduction in test RMSE is $3.48\%$, larger than the average reduction

rate of $1.70\%$ when $M$ increases from 2 to 4. While varying the number of views, we also monitor the changes in the model hyperparameters. Using the values that were selected to minimize the validation error of DiCoM-N, Table 2 shows the log ratio of $\log_{10}(\kappa_{\mathrm{div}}/\kappa_{\mathrm{csc}})$. For most datasets, we see that this log ratio tends to increase for larger number of views. This is because in $\mathfrak{L}_{\mathrm{DiCoM}}$, the number of diversity terms grows in $O(M)$ while the number of consistency terms grows in $O(M^2)$. Thus, as $M$ increases, a larger $(\kappa_{\mathrm{div}}/\kappa_{\mathrm{csc}})$ ratio is required to keep those terms balanced.

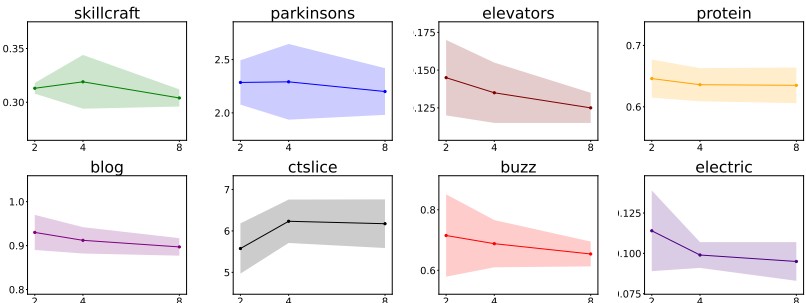

Figure 4: Test RMSE of DiCoM-N on UCI datasets with varying number of views $M \in \{2, 4, 8\}$. The x-axis shows number of views $M$, the y-axis shows test RMSE.

Table 2: Median Values of $\log_{10}(\kappa_{\mathrm{div}}/\kappa_{\mathrm{csc}})$ across 10 Seeds from DiCoM-N.

|          | skillcraft | parkinsons | elevators | protein | blog  | ctslice | buzz  | electric |
|----------|-----------|-----------|-----------|---------|-------|---------|-------|----------|
| $M = 2$  | 1.849     | $-0.151$  | 1.151     | 0.500   | 1.000 | $-1.000$| 0.151 | 1.849    |
| $M = 4$  | 2.000     | $-0.301$  | 1.151     | 0.301   | 1.151 | 0.199   | 0.349 | 2.000    |
| $M = 8$  | 1.699     | $-0.500$  | 1.500     | 1.151   | 1.849 | 0.349   | 0.500 | 2.000    |

**Comparing multi-network and multi-branch:** In this experiment, we compare the two variants of DiCoM against each other, by reporting both their performance and execution time (adding train and test time). Table 3 shows the percentage reduction computed using equation (11) by treating DiCoM-N's results as the baseline scores and DiCoM-B's corresponding results as the new scores. It can be seen that DiCoM-N is consistently outperforming DiCoM-B in terms of test RMSE. The multi-network variant is also the faster option when $M = 2$. However, as the number of views increases, the execution time of DiCoM-B is significantly faster. This shows that while DiCoM-N achieves better test performance, DiCoM-B demonstrates better scalability.

Table 3: From DiCoM-N to DiCoM-B: Percentage Reduction in Test RMSE and Execution Time.

| Dataset    | Test RMSE |         |         | Execution Time |         |         |
|------------|-----------|---------|---------|----------------|---------|---------|
|            | $M = 2$   | $M = 4$ | $M = 8$ | $M = 2$        | $M = 4$ | $M = 8$ |
| skillcraft | -0.702    | 2.012   | -2.945  | -33.336        | 23.505  | 48.462  |
| parkinsons | -43.903   | -24.177 | -57.821 | 36.845         | 55.590  | 72.481  |
| elevators  | -0.436    | -13.126 | -21.811 | -52.633        | 19.389  | 60.346  |
| protein    | 2.798     | -0.536  | -0.520  | -45.076        | 19.876  | 62.177  |
| blog       | -0.430    | -6.908  | -18.729 | -39.355        | 39.847  | 47.129  |
| ctslice    | -54.862   | -34.924 | -73.619 | 45.016         | 81.627  | 83.731  |
| buzz       | -15.885   | -4.984  | -17.976 | 4.194          | 33.759  | 51.035  |
| Average    | -16.203   | -11.806 | -27.632 | -12.049        | 39.085  | 60.766  |

## 4.2 EXPERIMENTS ON VISION DATA

In order to show the versatility of DiCoM, we further conduct experiments on a crowd counting task. Crowd counting is a fundamental question in the vision community due to its far-reaching applications in many scenarios, including video surveillance, metropolis security, human behavior analysis.

Crowd counting has been recently used as a benchmark for deep regression algorithms (Zhang et al., 2019); for this task, counting by regression has been perceived as the state-of-the-art approach.

**Dataset:** We study the ShanghaiTech Part-A dataset (Zhang et al., 2016)[2]. This is a new large-scale crowd counting dataset that contains extremely congested scenes, with varying perspective and unfixed resolution. The data are split into 300 training and 182 test samples. Among the 300 training samples, we randomly select $N \in \{30, 120, 210\}$ samples as the labeled set and use the remaining data as the unlabeled set. We follow the common practice to report both mean absolute errors (MAE) and root-mean-squared errors (RMSE) on the test set. We note that this dataset inevitably contains personally identifiable information, which has been made public by the owner of the dataset.

**Experiment Setup:** In our experiments, we adopt the network architecture of CSRNet B (Li et al., 2018) and implement DiCoM-B-4. More specifically, we use a pre-trained VGG16 network as the encoder and append another decoder on top of it. In the penultimate layer of the decoder, we enlarge the number of hidden channels by $M$ times. We then apply a group-convolutional layer as the last layer, setting both the number of output channels and group size to $M$. Thus, the backbone of DiCoM-B-4 includes the pre-trained VGG16 and the decoder up to its penultimate layer. This architecture has been shown to be very effective in generating multiple predictions without much increase in computational cost over a single network (Shi et al., 2018; Zhang et al., 2019; Zhou et al., 2021).

We compare DiCoM with the following competing methods: (i) the supervised baseline which uses only the labeled training samples and standard MSE loss; (ii) the DNCL model (Zhang et al., 2019); and (iii) the Co-Regression model. Since we are not running for multiple random seeds, we remove all random data augmentations (e.g., cropping, flipping) to enable fair comparison between different methods.

**Results:** From the results in Table 4, we observe that both DNCL and Co-Regression outperform the supervised baseline by enforcing either diversity on labeled data or consistency on unlabeled data, and that overall, DiCoM-B-4 outperforms other methods by incorporating both diversity and consistency on the unlabeled data.

Table 4: Test Results on ShanghaiTech.

|  | Supervised Baseline | | DNCL | | Co-Regression | | DiCoM-B-4 | |
|---|---|---|---|---|---|---|---|---|
|  | MAE | RMSE | MAE | RMSE | MAE | RMSE | MAE | RMSE |
| $N = 30$ | 297.45 | 551.68 | 250.91 | 360.66 | 224.63 | 344.46 | **163.41** | **227.42** |
| $N = 120$ | 132.05 | 210.82 | 119.48 | 186.80 | 128.57 | 206.90 | **114.31** | **181.91** |
| $N = 210$ | 107.76 | 165.91 | 104.00 | 165.97 | 103.20 | 165.04 | **101.90** | **155.49** |

## 5 CONCLUSION

In this paper, we proposed novel Diverse and Consistent Multi-view Networks for Semi-supervised Regression (DiCoM), that elegantly combines ensemble diversity with consistency regularization. DiCoM utilizes probabilistic graphical models to control the underlying dependencies among multiple regression outputs and label. We also show that DiCoM is highly flexible, it can be adopted for multi-network or multi-branch implementations, the latter significantly improves the scalability of the method. Experiments on tabular UCI and visual ShanghaiTech datasets demonstrated the effectiveness of the proposed method across diverse domains, while ablation studies validated the importance of both consistency and diversity. In the future, one may extend the DiCoM framework by introducing asymmetric views (of non-identical architectures), which naturally causes the final output $\mu$ to be an unequally-weighted average. Another interesting direction is to explore the potential impact of data augmentation techniques, since multi-view learning often benefits from diversified inputs.

---

[2]https://github.com/desenzhou/ShanghaiTechDataset (BSD 2-clause license)

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

## A MATHEMATICAL PROOFS

Consider again the general model, where there are $M$ deep views, i.e., $\{\mathbf{f}_m\}_{m=1}^M$. Graphically, these functions are represented by nodes that are connected not directly, but only via the consensus function $\mathbf{f}_c$ using isotropic Gaussian potentials

$$\mathbf{f}_c - \mathbf{f}_m \propto \mathcal{N}\left(\mathbf{0}, \sigma_m^2 \mathbf{I}\right). \tag{12}$$

We note that in this general case, each deep view is a vector (instead of a scalar) and is assigned a separate variance $\sigma_m^2$, which are not necessarily equal to each other. In terms of notation, we use italic letters for scalar variables and boldface letters for vectors and matrices.

### A.1 MARGINAL DENSITY OF THE VIEWS

In this proof, we derive the marginal distribution of the views. Given the DiCoM graphical model, it is necessary to integrate $\mathbf{f}_c$ out of the joint density distribution of the graph, because $\mathbf{f}_c$ is a latent variable. The joint density distribution function of this graphical model is as follows

$$p\left(\mathbf{f}_c, \mathbf{f}_1, \ldots, \mathbf{f}_M\right) = \frac{1}{\mathcal{Z}_1} \prod_{m=1}^M \exp\left(-\frac{\|\mathbf{f}_c - \mathbf{f}_m\|^2}{2\sigma_m^2}\right) \tag{13}$$

$$= \frac{1}{\mathcal{Z}_1} \exp\left(-\sum_m \frac{\mathbf{f}_c^\top \mathbf{f}_c}{2\sigma_m^2} + \sum_m \frac{\mathbf{f}_c^\top \mathbf{f}_m}{\sigma_m^2} - \sum_m \frac{\mathbf{f}_m^\top \mathbf{f}_m}{2\sigma_m^2}\right) \tag{14}$$

$$= \frac{1}{\mathcal{Z}_1} \exp\left(-\frac{\psi}{2}\mathbf{f}_c^\top \mathbf{f}_c + \boldsymbol{\phi}^\top \mathbf{f}_c + \chi\right), \tag{15}$$

where the normalizing factor $\mathcal{Z}_1$ is a constant w.r.t. $\mathbf{f}_c, \mathbf{f}_1, \ldots, \mathbf{f}_M$ and

$$\psi = \sum_{m=1}^M \frac{1}{\sigma_m^2} \qquad \boldsymbol{\phi} = \sum_{m=1}^M \frac{\mathbf{f}_m}{\sigma_m^2} \qquad \chi = \sum_{m=1}^M -\frac{\mathbf{f}_m^\top \mathbf{f}_m}{2\sigma_m^2}. \tag{16}$$

Notice that $\psi, \boldsymbol{\phi}, \chi$ are constants w.r.t. $\mathbf{f}_c$. By applying the following integration rule for a multivariate Gaussian variable $\mathbf{x}$ Petersen et al. (2008)

$$\int \exp\left(-\frac{1}{2}\mathbf{x}^\top \mathbf{A}\mathbf{x} + \mathbf{c}^\top \mathbf{x}\right) d\mathbf{x} = \sqrt{\det\left(2\pi \mathbf{A}^{-1}\right)} \exp\left(\frac{1}{2}\mathbf{c}^\top \mathbf{A}^{-\top}\mathbf{c}\right), \tag{17}$$

we can integrate $\mathbf{f}_c$ out of the joint distribution in (15) to obtain the following marginal likelihood

$$p\left(\mathbf{f}_1, \ldots, \mathbf{f}_M\right) = \int p\left(\mathbf{f}_c, \mathbf{f}_1, \ldots, \mathbf{f}_M\right) d\mathbf{f}_c \tag{18}$$

$$= \frac{1}{\mathcal{Z}_2} \exp\left(\frac{\boldsymbol{\phi}^\top \boldsymbol{\phi}}{2\psi} + \chi\right) \tag{19}$$

$$= \frac{1}{\mathcal{Z}_2} \exp\left[\frac{1}{2\psi}\left(\sum_m \frac{\mathbf{f}_m^\top \mathbf{f}_m}{\sigma_m^4} + 2\sum_m \sum_{k>m} \frac{\mathbf{f}_m^\top \mathbf{f}_k}{\sigma_m^2 \sigma_k^2} - \psi \sum_m \frac{\mathbf{f}_m^\top \mathbf{f}_m}{\sigma_m^2}\right)\right] \tag{20}$$

$$= \frac{1}{\mathcal{Z}_2} \exp\left[\frac{1}{2\psi}\left(\sum_m \sum_{k>m} -\frac{\mathbf{f}_m^\top \mathbf{f}_m - 2\mathbf{f}_m^\top \mathbf{f}_k + \mathbf{f}_k^\top \mathbf{f}_k}{\sigma_m^2 \sigma_k^2}\right)\right] \tag{21}$$

$$= \frac{1}{\mathcal{Z}_2} \exp\left(\sum_m \sum_{k>m} \frac{-\|\mathbf{f}_m - \mathbf{f}_k\|^2}{2\rho_{m,k}^2}\right) \tag{22}$$

$$= \frac{1}{\mathcal{Z}_2} \exp\left(\sum_m \sum_{k>m} -\lambda_{m,k}\|\mathbf{f}_m - \mathbf{f}_k\|^2\right), \tag{23}$$

where $\mathcal{Z}_2$ is another constant w.r.t. $\mathbf{f}_c, \mathbf{f}_1, \ldots, \mathbf{f}_M$, and

$$\rho_{m,k} = \sigma_m^2 \sigma_k^2 \psi = \sigma_m^2 \sigma_k^2 \left( \sum_{i=1}^{M} \frac{1}{\sigma_i^2} \right) \tag{24}$$

$$\lambda_{m,k} = (2\rho_{m,k})^{-1} = \left[ 2\sigma_m^2 \sigma_k^2 \left( \sum_{i=1}^{M} \frac{1}{\sigma_i^2} \right) \right]^{-1}. \tag{25}$$

## A.2 Conditional Density of the Consensus Function

In this proof, we derive the conditional density distribution of the consensus function $\mathbf{f}_c$ given the views. Consider again the general model with $M$ views, i.e., $\{\mathbf{f}_m\}_{m=1}^{M}$. Each view is represented by a random variable connected only to the consensus function $\mathbf{f}_c$ via an isotropic Gaussian potential as defined in (12). From (13), (22), the conditional distribution of $\mathbf{f}_c$ given the views $\mathbf{f}_1, \ldots, \mathbf{f}_M$ is

$$p\left(\mathbf{f}_c | \mathbf{f}_1, \ldots, \mathbf{f}_M\right) = \frac{p\left(\mathbf{f}_c, \mathbf{f}_1, \ldots \mathbf{f}_M\right)}{p\left(\mathbf{f}_1, \ldots, \mathbf{f}_M\right)} \tag{26}$$

$$= \frac{1}{\mathcal{Z}_3} \exp \left( \sum_{m=1}^{M} \frac{-\|\mathbf{f}_c - \mathbf{f}_m\|^2}{2\sigma_m^2} + \sum_{m=1}^{M} \sum_{k>m}^{M} \frac{\|\mathbf{f}_m - \mathbf{f}_k\|^2}{2\rho_{m,k}^2} \right) \tag{27}$$

$$= \frac{1}{\mathcal{Z}_3} \exp \left( \sum_{m} \frac{-\mathbf{f}_c^\top \mathbf{f}_c + 2\mathbf{f}_m^\top \mathbf{f}_c - \mathbf{f}_m^\top \mathbf{f}_m}{2\sigma_m^2} + \sum_{m} \sum_{k>m} \frac{\|\mathbf{f}_m - \mathbf{f}_k\|^2}{2\rho_{m,k}^2} \right) \tag{28}$$

$$= \frac{1}{\mathcal{Z}_3} \exp \left( \frac{-\mathbf{f}_c^\top \mathbf{f}_c}{2\sigma_\mu^2} + \frac{\tilde{\mu}^\top \mathbf{f}_c}{\sigma_\mu^2} + \aleph \right), \tag{29}$$

where the normalizing factor $\mathcal{Z}_3$ is a constant w.r.t. $\mathbf{f}_c, \mathbf{f}_1, \ldots, \mathbf{f}_M$ and

$$\sigma_\mu^2 = \left( \sum_{m=1}^{M} \frac{1}{\sigma_m^2} \right)^{-1} \tag{30}$$

$$\tilde{\mu} = \sigma_\mu^2 \sum_{m=1}^{M} \frac{\mathbf{f}_m}{\sigma_m^2} \tag{31}$$

$$\aleph = \sum_{m=1}^{M} \frac{-\mathbf{f}_m^\top \mathbf{f}_m}{2\sigma_m^2} + \sum_{m=1}^{M} \sum_{k>m}^{M} \frac{\|\mathbf{f}_m - \mathbf{f}_k\|^2}{2\rho_{m,k}^2}. \tag{32}$$

Using the definitions (24), (30) and (31), $\aleph$ can be rewritten as follows

$$\aleph = \sigma_\mu^2 \left( \sum_{m} \frac{-\mathbf{f}_m^\top \mathbf{f}_m}{2\sigma_m^2 \sigma_\mu^2} + \sum_{m} \sum_{k>m} \frac{\|\mathbf{f}_m - \mathbf{f}_k\|^2}{2\sigma_m^2 \sigma_k^2} \right) \tag{33}$$

$$= \sigma_\mu^2 \left[ \left( \sum_{m} \frac{-\mathbf{f}_m^\top \mathbf{f}_m}{2\sigma_m^2} \right) \left( \sum_{k} \frac{1}{\sigma_k^2} \right) + \sum_{m} \sum_{k>m} \frac{\|\mathbf{f}_m - \mathbf{f}_k\|^2}{2\sigma_m^2 \sigma_k^2} \right] \tag{34}$$

$$= \sigma_\mu^2 \left( \sum_{m} \sum_{k} \frac{-\mathbf{f}_m^\top \mathbf{f}_m}{2\sigma_m^2 \sigma_k^2} + \sum_{m} \sum_{k \neq m} \frac{\mathbf{f}_m^\top \mathbf{f}_m}{2\sigma_m^2 \sigma_k^2} + \sum_{m} \sum_{k>m} \frac{-\mathbf{f}_m^\top \mathbf{f}_k}{\sigma_m^2 \sigma_k^2} \right) \tag{35}$$

$$= \sigma_\mu^2 \left( \sum_{m} \frac{-\mathbf{f}_m^\top \mathbf{f}_m}{2\sigma_m^4} + \sum_{m} \sum_{k>m} \frac{-\mathbf{f}_m^\top \mathbf{f}_k}{\sigma_m^2 \sigma_k^2} \right) \tag{36}$$

$$= -\frac{\sigma_\mu^2}{2} \left( \sum_{m} \frac{\mathbf{f}_m^\top}{\sigma_m^2} \right) \left( \sum_{m} \frac{\mathbf{f}_m}{\sigma_m^2} \right) \tag{37}$$

$$= \frac{-\tilde{\mu}^\top \tilde{\mu}}{2\sigma_\mu^2}. \tag{38}$$

Thus, we can rewrite (29) in its Gaussian form

$$p\left(\mathbf{f}_c|\mathbf{f}_1,\ldots,\mathbf{f}_M\right) = \frac{1}{\mathcal{Z}_3}\exp\left(\frac{-\mathbf{f}_c^\top\mathbf{f}_c}{2\sigma_\mu^2} + \frac{\tilde{\mu}^\top\mathbf{f}_c}{\sigma_\mu^2} - \frac{\tilde{\mu}^\top\tilde{\mu}}{2\sigma_\mu^2}\right) = \frac{1}{\mathcal{Z}_3}\exp\left(-\frac{\|\mathbf{f}_c - \tilde{\mu}\|^2}{2\sigma_\mu^2}\right). \tag{39}$$

Therefore,

$$\mathbf{f}_c|\mathbf{f}_1,\ldots,\mathbf{f}_M \sim \mathcal{N}\left(\tilde{\mu}, \sigma_\mu^2\mathbf{I}\right). \tag{40}$$

## B  EXPERIMENT SETUP DETAILS

In this section, we provide the detailed setups for our benchmarking experiments. All experiments are run on NVIDIA GeForce GTX 1080Ti GPUs, using an Anaconda virtual environment installed with CUDA version 10.1, Python version 3.7.10 and Pytorch version 1.7.0.

**Experiment setup for DiCoM in UCI experiments:**

- Network architecture: for DiCoM-N, fully-connected multilayer perceptron with hidden layers of size $[100, 50, 50, 2]$. For DiCoM-B, the backbone includes the first 3 hidden layers of size $[100, 50, 50]$ and the branches include one hidden layer of size 2. Note that we are not counting the input and output layers.

- Parallelization: for DiCoM-B, we use group convolution in order to back-propagate through all branches simultaneously.

- Random seeds: $20, 40, \dots, 200$ (10 seeds in total).

- Training: 2000 epochs with 250 epoch patience for early stopping (stop if no improvement is observed on validation set for 250 consecutive epochs).

- Optimizer: Stochastic Gradient Descent with momentum 0.95 and weight decay $10^{-9}$. Learning rate is $10^{-4}$ for ctslice and is $10^{-3}$ for other datasets.

- Augmentation: additive random Gaussian noise with mean 0 and standard deviation 0.05 for DiCoM-N and 0.01 for DiCoM-B.

- Diversity hyperparameter search range: $\kappa_{\text{div}} \in \{0.01, 0.05, 0.1, 0.5, 1\}$.

- Consistency hyperparameter search range: $\kappa_{\text{csc}} \in \{0.01, 0.05, 0.1, 0.5, 1\}$.

**Experiment setup for DiCoM in ShanghaiTech experiment:**

- Network architecture: CSRNet B Li et al. (2018).

- Random seeds: 9999 (only 1 seed).

- Training: 1000 epochs with no early stopping.

- Optimizer: Adam with weight decay $10^{-5}$ and learning rate $10^{-5}$.

- Augmentation: None.

- Diversity hyperparameter search range: $\kappa_{\text{div}} \in \{10^{-5}, 10^{-4}, 10^{-3}\}$.

- Consistency hyperparameter search range: $\kappa_{\text{csc}} \in \{10^{-5}, 10^{-4}, 10^{-3}\}$.

## C   ADDITIONAL EXPERIMENT RESULTS ON UCI DATASETS

**Information of UCI datasets:** The detailed statistics of the eight UCI datasets are given in the table below.

Table 5: UCI Regression Datasets

| Dataset | No. of Samples | Input Dim. | No. of Unique Label Values | Prediction Target |
|---|---|---|---|---|
| **skillcraft** | 3,325 | 18 | 7 | Skill level of gamers (ordinal classification) |
| **parkinsons** | 5,875 | 20 | 1,129 | Unified Parkinson's Disease Rating Scale (UPDRS) scores |
| **elevators** | 16,599 | 18 | 61 | Aileron control of F16 aircraft |
| **protein** | 45,730 | 9 | 15,903 | Physicochemical properties of protein tertiary structure |
| **blog** | 52,397 | 280 | 438 | Number of comments received within 24 hrs |
| **ctslice** | 53,500 | 384 | 53,347 | Relative location of the image on the axial axis |
| **buzz** | 583,250 | 77 | 8,123 | Popularity of a topic in social media |
| **electric** | 2,049,280 | 6 | 4,186 | Power consumption in one household per minute |

**Additional experiment results on UCI datasets with labeling budget** $N = 300$**:** Please see the below additional results from Section 4.1.

Table 6: Test RMSE on UCI Datasets with Labeling Budget $N = 300$.

| Dataset | SSDKL | COREG | LP | VAE | MT | VAT | DiCoM-N-2 | DiCoM-B-2 |
|---|---|---|---|---|---|---|---|---|
| skillcraft | $0.346 \pm 0.036$ | $0.361 \pm 0.009$ | $0.342 \pm 0.012$ | $0.396 \pm 0.014$ | $0.418 \pm 0.038$ | $0.454 \pm 0.025$ | $\mathbf{0.313 \pm 0.005}$ | $0.315 \pm 0.013$ |
| parkinsons | $3.379 \pm 0.387$ | $4.285 \pm 0.200$ | $5.371 \pm 0.994$ | $10.655 \pm 0.151$ | $8.725 \pm 0.904$ | $10.612 \pm 0.198$ | $\mathbf{2.285 \pm 0.208}$ | $3.289 \pm 0.481$ |
| elevators | $0.151 \pm 0.013$ | $0.195 \pm 0.006$ | $0.189 \pm 0.005$ | $0.228 \pm 0.010$ | $0.261 \pm 0.031$ | $0.261 \pm 0.012$ | $\mathbf{0.145 \pm 0.025}$ | $0.146 \pm 0.035$ |
| protein | $0.714 \pm 0.043$ | $0.703 \pm 0.020$ | $0.685 \pm 0.011$ | $0.780 \pm 0.017$ | $0.763 \pm 0.018$ | $0.766 \pm 0.010$ | $0.646 \pm 0.031$ | $\mathbf{0.628 \pm 0.014}$ |
| blog | $1.146 \pm 0.072$ | $1.076 \pm 0.040$ | $1.017 \pm 0.048$ | $1.033 \pm 0.045$ | $1.117 \pm 0.036$ | $1.087 \pm 0.063$ | $\mathbf{0.930 \pm 0.040}$ | $0.934 \pm 0.036$ |
| ctslice | $9.691 \pm 1.135$ | $10.023 \pm 1.235$ | $10.230 \pm 1.410$ | $16.301 \pm 0.773$ | $16.845 \pm 1.996$ | $16.992 \pm 4.839$ | $\mathbf{5.575 \pm 0.606}$ | $8.634 \pm 0.923$ |
| buzz | $0.766 \pm 0.040$ | $0.752 \pm 0.035$ | $0.851 \pm 0.030$ | $1.098 \pm 0.026$ | $1.702 \pm 0.160$ | $1.743 \pm 0.186$ | $\mathbf{0.715 \pm 0.136}$ | $0.829 \pm 0.065$ |
| electric | $0.117 \pm 0.011$ | $0.261 \pm 0.008$ | $0.418 \pm 0.010$ | $0.977 \pm 0.017$ | $0.696 \pm 0.023$ | $0.916 \pm 0.127$ | $\mathbf{0.114 \pm 0.025}$ | $0.279 \pm 0.021$ |

Table 7: Test RMSE of DiCoM-N on UCI Datasets with Varying Number of Views, $N = 300$

| Dataset | DiCoM-N-2 | DiCoM-N-4 | | DiCoM-N-8 | |
|---|---|---|---|---|---|
| | RMSE | RMSE | % Redc. $2 \to 4$ | RMSE | % Redc. $4 \to 8$ |
| **skillcraft** | $0.313 \pm 0.005$ | $0.319 \pm 0.025$ | -1.917 | $\mathbf{0.304 \pm 0.008}$ | 4.702 |
| **parkinsons** | $2.285 \pm 0.208$ | $2.291 \pm 0.355$ | -0.263 | $\mathbf{2.200 \pm 0.219}$ | 3.972 |
| **elevators** | $0.145 \pm 0.025$ | $0.135 \pm 0.020$ | 7.187 | $\mathbf{0.125 \pm 0.010}$ | 7.407 |
| **protein** | $0.646 \pm 0.031$ | $0.636 \pm 0.027$ | 1.548 | $\mathbf{0.635 \pm 0.029}$ | 0.157 |
| **blog** | $0.930 \pm 0.040$ | $0.912 \pm 0.030$ | 1.935 | $\mathbf{0.897 \pm 0.020}$ | 1.645 |
| **ctslice** | $\mathbf{5.575 \pm 0.606}$ | $6.233 \pm 0.524$ | -11.795 | $6.174 \pm 0.587$ | 0.947 |
| **buzz** | $0.715 \pm 0.136$ | $0.688 \pm 0.078$ | 3.804 | $\mathbf{0.654 \pm 0.041}$ | 4.942 |
| **electric** | $0.114 \pm 0.025$ | $0.099 \pm 0.008$ | 13.132 | $\mathbf{0.095 \pm 0.012}$ | 4.058 |
| **Average** | | | 1.704 | | 3.479 |

**Additional experiment results on UCI datasets with labeling budget** $N = 100$**:** Using the same UCI datasets, we conduct experiments similar to the ones in Section 4.1 with a smaller labeling budget of $N = 100$ samples. The results are provided below.

Table 8: Test RMSE on UCI Datasets with Labeling Budget $N = 100$.

| Dataset | SSDKL | COREG | LP | VAE | MT | VAT | DiCoM-N-2 | DiCoM-B-2 |
|---|---|---|---|---|---|---|---|---|
| skillcraft | $0.360 \pm 0.029$ | $0.360 \pm 0.013$ | $0.345 \pm 0.016$ | $0.412 \pm 0.020$ | $0.447 \pm 0.039$ | $0.454 \pm 0.034$ | $0.342 \pm 0.011$ | $\mathbf{0.330 \pm 0.021}$ |
| parkinsons | $4.701 \pm 0.687$ | $5.552 \pm 0.358$ | $7.037 \pm 1.373$ | $10.694 \pm 0.216$ | $8.998 \pm 0.778$ | $10.738 \pm 0.254$ | $\mathbf{3.580 \pm 0.662}$ | $4.442 \pm 0.929$ |
| elevators | $0.182 \pm 0.019$ | $0.213 \pm 0.009$ | $0.202 \pm 0.008$ | $0.251 \pm 0.029$ | $0.269 \pm 0.039$ | $0.261 \pm 0.015$ | $\mathbf{0.179 \pm 0.030}$ | $0.194 \pm 0.029$ |
| protein | $0.741 \pm 0.050$ | $0.751 \pm 0.015$ | $0.746 \pm 0.022$ | $0.789 \pm 0.036$ | $0.757 \pm 0.025$ | $0.784 \pm 0.050$ | $0.691 \pm 0.021$ | $\mathbf{0.675 \pm 0.013}$ |
| blog | $1.129 \pm 0.081$ | $1.067 \pm 0.078$ | $1.115 \pm 0.061$ | $1.051 \pm 0.102$ | $1.112 \pm 0.053$ | $1.136 \pm 0.111$ | $1.031 \pm 0.055$ | $\mathbf{0.998 \pm 0.052}$ |
| ctslice | $12.646 \pm 1.059$ | $12.911 \pm 0.983$ | $13.726 \pm 1.178$ | $19.750 \pm 3.178$ | $17.752 \pm 1.352$ | $18.318 \pm 3.720$ | $\mathbf{10.742 \pm 0.629}$ | $12.062 \pm 1.413$ |
| buzz | $0.904 \pm 0.117$ | $\mathbf{0.830 \pm 0.063}$ | $0.911 \pm 0.066$ | $1.114 \pm 0.036$ | $1.786 \pm 0.125$ | $1.625 \pm 0.339$ | $0.836 \pm 0.093$ | $0.974 \pm 0.059$ |
| electric | $\mathbf{0.152 \pm 0.030}$ | $0.352 \pm 0.020$ | $0.484 \pm 0.032$ | $0.982 \pm 0.034$ | $0.838 \pm 0.105$ | $0.937 \pm 0.071$ | $0.185 \pm 0.116$ | $0.308 \pm 0.043$ |

Table 9: Test RMSE from ablation study on UCI datasets, $N = 100$

| | DiCoM-N-2 | Ablation-1 | | Ablation-2 | | Ablation-3 | | Ablation-4 | |
|---|---|---|---|---|---|---|---|---|---|
| **Aug.** | ✓ | | | ✓ | | ✓ | | ✓ | |
| **Lab. Div.** | ✓ | ✓ | | | | ✓ | | ✓ | |
| **Unlab. Csc.** | ✓ | ✓ | | ✓ | | | | | |
| **Unlab. Div.** | | | | | | | | ✓ | |
| | RMSE | RMSE | % Redc. | RMSE | % Redc. | RMSE | % Redc. | RMSE | % Redc. |
| **skillcraft** | **0.342 ± 0.011** | 0.354 ± 0.030 | -3.404 | 0.361 ± 0.032 | -5.372 | 0.355 ± 0.031 | -3.756 | 0.363 ± 0.032 | -6.124 |
| **parkinsons** | **3.580 ± 0.662** | 4.044 ± 0.623 | -12.969 | 3.688 ± 0.674 | -3.039 | 3.674 ± 0.326 | -2.643 | 3.516 ± 0.374 | 1.783 |
| **elevators** | **0.179 ± 0.030** | 0.186 ± 0.033 | -3.844 | 0.192 ± 0.032 | -6.864 | 0.193 ± 0.029 | -7.574 | 0.197 ± 0.026 | -9.775 |
| **protein** | **0.691 ± 0.021** | 0.705 ± 0.023 | -1.959 | 0.711 ± 0.013 | -2.818 | 0.723 ± 0.040 | -4.603 | 0.714 ± 0.023 | -3.268 |
| **blog** | **1.031 ± 0.055** | 1.074 ± 0.052 | -4.214 | 1.090 ± 0.043 | -5.721 | 1.124 ± 0.072 | -9.030 | 1.117 ± 0.081 | -8.295 |
| **ctslice** | 10.742 ± 0.629 | 10.822 ± 0.564 | -0.747 | 11.341 ± 1.114 | -5.575 | **10.290 ± 0.911** | 4.209 | 10.673 ± 1.070 | 0.642 |
| **buzz** | **0.836 ± 0.093** | 0.918 ± 0.085 | -9.760 | 0.933 ± 0.263 | -11.599 | 1.019 ± 0.183 | -21.893 | 1.017 ± 0.242 | -21.613 |
| **electric** | 0.185 ± 0.116 | 0.176 ± 0.103 | 4.602 | 0.198 ± 0.115 | -6.798 | **0.158 ± 0.023** | 14.693 | 0.193 ± 0.104 | -4.460 |
| **Average** | | | -4.037 | | -5.973 | | -3.825 | | -6.389 |

Table 10: Test RMSE of DiCoM-N on UCI Datasets with Varying Number of Views, $N = 100$

| Dataset | DiCoM-N-2 | DiCoM-N-4 | | DiCoM-N-8 | |
|---|---|---|---|---|---|
| | RMSE | RMSE | % Redc. $2 \rightarrow 4$ | RMSE | % Redc. $4 \rightarrow 8$ |
| **skillcraft** | 0.342 ± 0.011 | 0.349 ± 0.022 | -1.977 | **0.334 ± 0.018** | 4.250 |
| **parkinsons** | 3.580 ± 0.662 | 3.343 ± 0.394 | 6.610 | **3.250 ± 0.323** | 2.782 |
| **elevators** | 0.179 ± 0.030 | 0.171 ± 0.028 | 4.605 | **0.169 ± 0.028** | 1.185 |
| **protein** | 0.691 ± 0.021 | 0.681 ± 0.021 | 1.456 | **0.679 ± 0.024** | 0.294 |
| **blog** | 1.031 ± 0.055 | 0.975 ± 0.038 | 5.432 | **0.961 ± 0.053** | 1.436 |
| **ctslice** | 10.742 ± 0.629 | 11.164 ± 1.972 | -3.924 | **10.457 ± 1.109** | 6.333 |
| **buzz** | 0.836 ± 0.093 | 0.832 ± 0.083 | 0.447 | **0.796 ± 0.052** | 4.350 |
| **electric** | 0.185 ± 0.116 | 0.171 ± 0.059 | 7.650 | **0.137 ± 0.029** | 20.096 |
| **Average** | | | 2.537 | | 5.091 |

Table 11: From DiCoM-N to DiCoM-B: Percentage Reduction in Test RMSE and Execution Time, $N = 100$.

| Dataset | Test RMSE | | | Execution Time | | |
|---|---|---|---|---|---|---|
| | $M = 2$ | $M = 4$ | $M = 8$ | $M = 2$ | $M = 4$ | $M = 8$ |
| **skillcraft** | 3.503 | -1.323 | -10.419 | -72.882 | 18.200 | 59.470 |
| **parkinsons** | -24.102 | -30.058 | -57.663 | 28.085 | 28.085 | 65.291 |
| **elevators** | -8.069 | -13.635 | -20.037 | -11.291 | 37.481 | 37.481 |
| **protein** | 2.260 | -5.401 | -15.882 | -57.697 | 20.049 | 63.618 |
| **blog** | 3.219 | -4.069 | -11.128 | 23.903 | 23.812 | 56.539 |
| **ctslice** | -7.157 | -6.599 | -45.748 | 22.144 | 42.888 | 77.162 |
| **buzz** | -16.619 | -9.784 | -25.956 | 0.524 | 46.706 | 68.124 |
| **Average** | -6.709 | -10.124 | -26.690 | -9.602 | 31.032 | 61.098 |

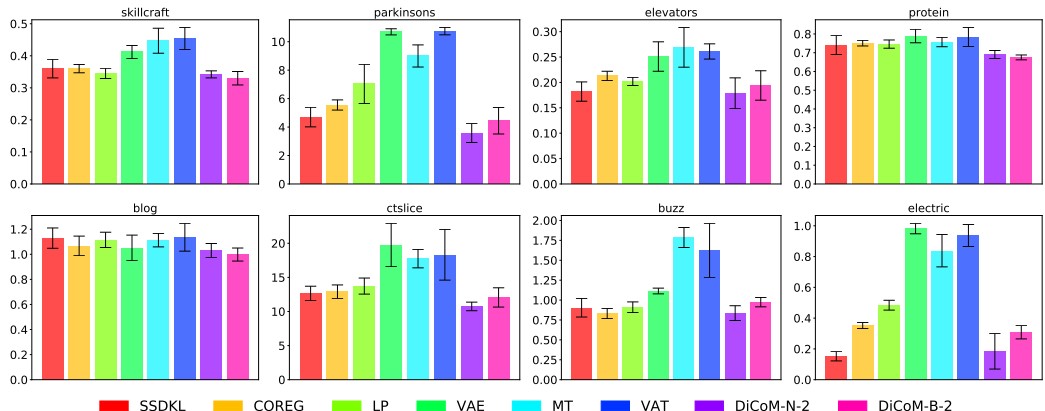

Figure 5: Test RMSE on UCI datasets: each subplot shows the results for one dataset, $N = 100$.

## D  ADDITIONAL EXPERIMENT ON TOY DATA

We conduct an experiment on a synthetic toy dataset to illustrate how multi-view diversity and consistency work together to affect the training and inference of DiCoM-N.

**Dataset:** We synthesize a regression dataset where inputs $x \in \mathbb{R}^{30}$ and labels $y \in \mathbb{R}^2$. The labels are related to the inputs by $y = Ax + \epsilon$, where $A$ is a fixed $2 \times 30$ coefficient matrix and $\epsilon \sim \mathcal{N}(\mathbf{0}, 0.3^2\mathbf{I})$. Each coordinate of $x$ is drawn from the standard normal distribution. We generate a training set of 100 labeled and 1000 unlabeled samples, and a hold-out test set of 1000 labeled samples.

**Experiment Setup:** Our DiCoM-N model has $M = 5$ views, each uses a simple neural network with a single hidden layer containing two hidden nodes. We train the model with SGD for 50 epochs with a learning rate of $5 \times 10^{-2}$, then evaluate the mean-squared-error (MSE) of the model on the test set.

**Results:** We keep $\kappa_{\mathrm{div}} = 1$ and vary $\kappa_{\mathrm{csc}}$ on a log scale: $\kappa_{\mathrm{csc}} \in \{0.01, 0.1, 1\}$. Both quantitative and qualitative results are shown in Fig. 6. We plot the training losses on the top row and visually show the predictions of each network on eight random test samples. On the left scenario (Fig. 6(a)) when $\kappa_{\mathrm{div}} \gg \kappa_{\mathrm{csc}}$, the diversity loss dominates the consistency component. Even though the total loss converges on the training set, the individual views' losses do not, resulting in their large bias on test samples. This can be an issue if the DiCoM-N model contains a smaller number of views. On the other hand, the consistency enforcement is too strong on the right scenario (Fig. 6(c)). The individual views and the averaged output seem to have all collapsed into a single point. Where the models collapse to is limited by the individual networks' capacity and may not necessarily be the global optimum for the averaged output. Finally, in the middle scenario (Fig. 6(b)), the effects of diversity and consistency losses are balanced, yielding a good trade-off. The averaged model output is able to perform better than each individual view, and is also the best among three scenarios. These results also suggest that even though diversity and consistency are contradicting forces, they can still be applied simultaneously on the regression outputs to produce the desirable behaviours.

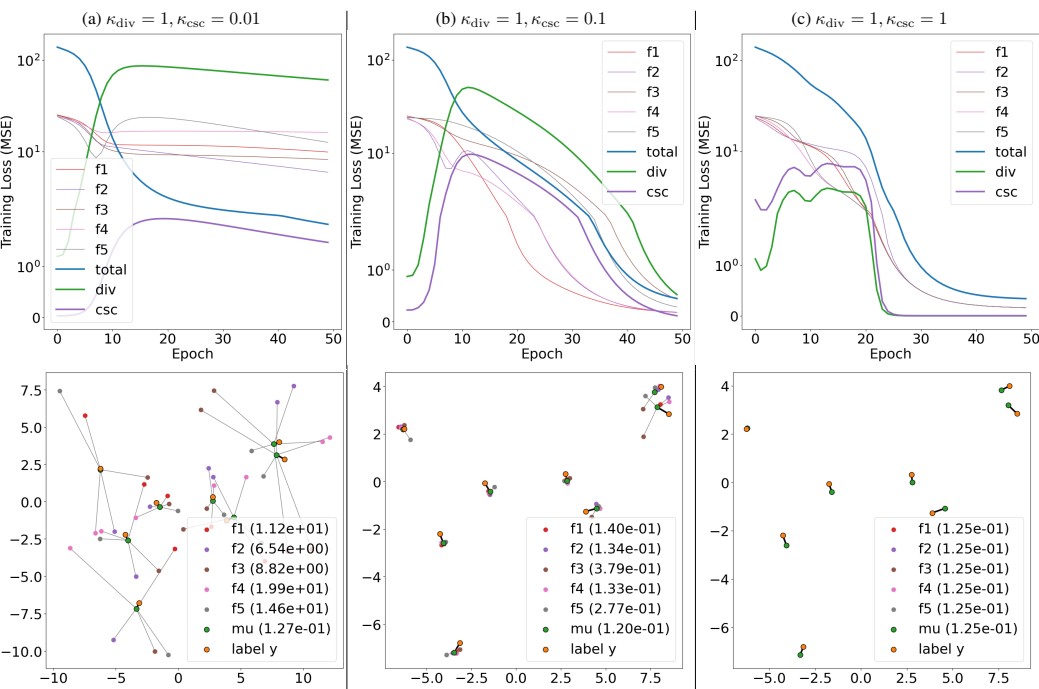

Figure 6: Experiment results on toy data. The top row shows training losses in symmetric log scale. The bottom row shows model predictions on eight random test samples. In the legend, next to the model name, we report the MSE scores evaluated on 1000 test samples.

