# OpenReview forum: "Diverse and Consistent Multi-view Networks for Semi-supervised Regression"
_ICLR.cc/2022/Conference — ICLR 2022 Submitted_

### Official Review · Reviewer_C8H3 · 2021-10-28

**Correctness:** 4
**Technical Novelty And Significance:** 2
**Empirical Novelty And Significance:** 3
**Recommendation:** 6
**Confidence:** 3

**Main Review:**

## Strengths

This paper studied the semi-supervised regression problem and tackled it with a technically sound approach.
It sheds light on the effects of diversity and consistency in multi-view learning.
Besides, this paper is mostly clear written.
It is easy to follow the derivation and verify its correctness.

---

## Weaknesses

My biggest concern is its novelty.
As cited in the paper, most of the probabilistic graphical model definition, marginalization, and likelihood derivation is quite similar to those used in [Yu et al., 2011].
Although [Yu et al., 2011] used the Gaussian process while this work mainly used neural network and gradient-based optimization, the techniques for deriving the learning objective are basically the same.

Concretely, if I didn't misunderstand anything, the definition of the undirected graphical model is roughly as follows:

- $p(y, f_c, f_1, \dots, f_M, x) = p(y, f_c) \prod_{m=1}^M p(f_c, f_m) p(f_m, x)$

- $p(f_m, x) \propto F_m(\eta_m(x), \theta_m)$ (multi-view neural networks)

- $p(f_c, f_m) \propto \exp\left(-\frac{(f_c - f_m)^2}{2\sigma^2_m}\right)$ (consensus potential, "within-view potential" $p(f_m)$ in [Yu et al., 2011] does not appear here)

- $p(y, f_c) \propto \exp\left(-\frac{(y - f_c)^2}{2\sigma^2}\right)$ (output potential, Gaussian noise model for regression)

Then, both [Yu et al., 2011] and this work marginalize the potential function to obtain:

- $p(f_1, \dots, f_M) \propto \int \prod_{m=1}^M p(f_c, f_m) \,\mathrm{d} f_c$ (similar to [Yu et al., 2011] Eq. (7) in Section 3.1 (Marginal 1) and Appendix A.1 (derivation) without the within-view potential)

And the conditional probability is calculated via Bayes' rule:

- $p(f_c | f_1, \dots, f_M) = \frac{p(f_c, f_1, \dots, f_M)}{p(f_1, \dots, f_M)} \propto \exp\left(-\sum_{m=1}^M \frac{(f_c - f_m)^2}{2\sigma^2_m}  + \sum_{m<k} \lambda_{m,k} (f_m - f_k)^2\right)$

It seems that [Yu et al., 2011] can be used for semi-supervised multi-view regression as well as classification.
I'd like to know the main challenge and unique contribution of this work compared with [Yu et al., 2011].

---

## Questions and comments

### Regression

- I might have missed something, but is there anything "regression-specific" in this work?
  If we just change the form of likelihood, can we apply the same technique to classification as well?

### Data augmentation

- How do you perform data augmentation on tabular data?
- How do you perform data augmentation after the backbone network in DiCoM-B?
- The data augmentation seems non-differentiable, so the backbone is pre-trained and fixed?

### Graphical model notation

- Figure 2 does not seem to be standard probabilistic graphical model notation so it's a little confusing.
  For example, edges should not be labeled and $\sigma^2$ and $\lambda$ should be fixed parameters.

### Others

- Eq. (1): Shouldn't it be $\sigma^2_m$? The assumption of $\sigma^2_m = \sigma^2_v$ is not given here.
- Eq. (2): $p(f_m, f_c)$ is not defined nor explained.

---

## References

- Yu, Shipeng, et al. "Bayesian co-training." The Journal of Machine Learning Research 12 (2011): 2649-2680.

**Summary Of The Paper:**

This paper proposed a method for **semi-supervised multi-view regression** based on **data augmentation** and an **undirected graphical model**.
The author derived a "**consistency**" term for unlabeled data and a "**diversity**" term for labeled data based on their log-likelihood and combined them linearly with two hyperparameters.
The proposed method was evaluated on tabular and image data and analyzed via ablation study.

**Summary Of The Review:**

This paper proposed a technically sound method based on an undirected graphical model for semi-supervised regression.
However, compared to an existing formulation [Yu et al., 2011], its novelty and contribution are a little questionable and need to be further highlighted.
Hence, I think it's on the borderline for now.

---

> ### Author Response · Authors · 2021-11-17
> **Response to Reviewer C8H3 (part 1/2)**
>
> We thank the reviewer for carefully reading our paper and for giving constructive comments. Our responses to your comments are given below.
>
> > My biggest concern is its novelty. As cited in the paper, most of the probabilistic graphical model definition, marginalization, and likelihood derivation is quite similar to those used in [Yu et al., 2011]. Although [Yu et al., 2011] used the Gaussian process while this work mainly used neural network and gradient-based optimization, the techniques for deriving the learning objective are basically the same.
> ...
> It seems that [Yu et al., 2011] can be used for semi-supervised multi-view regression as well as classification. I'd like to know the main challenge and unique contribution of this work compared with [Yu et al., 2011].
>
> We would like to point out the fundamental differences between our DiCoM method and the Bayesian Co-training method (BCT) [Yu et al., 2011]:
> * Conceptually, BCT does not promote diversity among different views. It uses the consensus potential to fuse multiple kernels from different views, without modeling how the views can be made diverse from each other. On the other hand, DiCoM encourages the errors of the views to be negatively correlated, thus enforcing diversity among the views themselves.
> * Technically, BCT employs Gaussian Process (GP) as its base learner. The multi-view fusion is achieved by computing a co-training kernel from multiple single-view kernels. If the BCT authors want to enforce diversity among the views, the regularization would need to be applied on the GP kernels. For DiCoM, we do not assume GP priors on the views and derive a loss function from this modified graphical model. This makes DiCoM more generic (no longer constrained by the GP assumption) and allows end-to-end training via backpropagation. As a result, the DiCoM method is compatible with deep learning.
>
> Thus, with the DiCoM model, we have made fundamental conceptual and technical improvements compared to the BCT model [Yu et al., 2011].

---

> ### Author Response · Authors · 2021-11-17
> **Response to Reviewer C8H3 (part 2/2)**
>
> > Regression
> > * I might have missed something, but is there anything "regression-specific" in this work? If we just change the form of likelihood, can we apply the same technique to classification as well?
>
> We thank the reviewer for bringing this up. In order to make DiCoM compatible with classification, we believe it is not as simple as changing the form of likelihood due to the following reasons. First, the diversity term in our DiCoM loss function uses negative correlation, which is not applicable to classification in general. One may explore other diversity-promoting tools such as Determinantal Point Process [1] which work for classification. Second, the mathematical derivations in DiCoM are based on a fundamental isotropic Gaussian potential function (see equation (1)). This potential does not apply to classification in general (except for binary classification). Thus, our method is only proposed for the regression task.
>
> [1] Kulesza, Alex, and Ben Taskar. "Determinantal Point Processes for Machine Learning." Foundations and Trends® in Machine Learning 5.2–3 (2012): 123-286.
>
>
> > Data augmentation
> > * How do you perform data augmentation on tabular data?
> > * How do you perform data augmentation after the backbone network in DiCoM-B?
> > * The data augmentation seems non-differentiable, so the backbone is pre-trained and fixed?
>
> We thank the reviewer for these good questions. In our experiments on tabular data (UCI datasets), we only use zero-mean Gaussian noise which is commonly used for tabular data. For the DiCoM-B model, Gaussian noise is applied on the input and on the features at the beginning of each branch, right after branching out. Since the independent Gaussian noise is added during the forward pass, it does not affect the gradient values during backpropagation. The DiCoM-B model is trained end-to-end, the backbone is trained together with the branches. These details have been added to the Experiment section of Revision-1.
>
>
> > Graphical model notation
> > * Figure 2 does not seem to be standard probabilistic graphical model notation so it's a little confusing. For example, edges should not be labeled and $\sigma^2$ and $\lambda$ should be fixed parameters.
>
> We thank the reviewer for this helpful comment. We have made several changes to Figure 2 according to your suggestions. The annotations on the edges have been removed and the edge potential functions have been added to the bottom of the figure.
>
>
> > Others
> > * Eq. (1): Shouldn't it be $\sigma_m^2$? The assumption of $\sigma_m^2=\sigma_v^2$ is not given here.
>
> We thank the reviewer for carefully reading our paper. Yes, the variance should be $\sigma_m^2$ in equation (1). We have fixed this in Revision-1.
>
> > * Eq. (2): $p(f_m,f_c)$ is not defined nor explained.
>
> We thank the reviewer for bringing this up. In equation (2), $p(f_m,f_ c)$ was intended to denote the Gaussian potential between two nodes $f_c$ and $f_m$. In Revision-1, we have used $\mathbf{\Psi}(f_c,f_m)$ to properly denote the potential function.

---

> > ### Comment · Reviewer_C8H3 · 2021-11-30
> > **Post-rebuttal**
> >
> > I've checked the revised paper and the author's response to all reviews. I appreciate the author's response, which answered some of my questions well. Therefore I would like to increase my rating from 5 to 6. But I couldn't go higher than that based on the current manuscript and its contribution. For example, I'm afraid that I wouldn't say the difference between BCT and DiCoM is "fundamental" just because they use different base learners while using very similar undirected graphical models. Further, it is not that deep learning is always suitable for every situation. For tabular data, GP or some tree-based models may also perform well. It is nice that DiCoM can be used for high-dimensional data (Section 4.2) and it would be great if the author can further investigate it and highlight this contribution.

---

> > > ### Author Response · Authors · 2021-12-03
> > > **Response to Reviewer C8H3**
> > >
> > > We thank the reviewer for carefully reading all our responses and for giving us very helpful comments.
> > >
> > > Regarding the following suggestion
> > > >It is nice that DiCoM can be used for high-dimensional data (Section 4.2) and it would be great if the author can further investigate it and highlight this contribution.
> > >
> > > We will definitely consider expanding Section 4.2 in a future revision. Since DiCoM is a generic semi-supervised regression method, we also expect it to yield good performance on different data modalities and/or regression tasks.
> > >
> > > Regarding the following concern
> > > > For example, I'm afraid that I wouldn't say the difference between BCT and DiCoM is "fundamental" just because they use different base learners while using very similar undirected graphical models.
> > >
> > > We kindly remind the reviewer that DiCoM is *conceptually different* from BCT by promoting multi-view diversity. BCT does not learn diversity. Instead, it requires *physically-distinct views* (e.g., image and text describing the same object are two physical views) as input and assumes that the views are conditionally independent given the label. If these conditions are not satisfied, BCT performs worse than supervised learning (see the 4th paragraph of Section 6.2 of the BCT paper). On the other hand, DiCoM does not suffer from such restrictive conditions. In fact, DiCoM operates on *deep views that are generated from the same input*, i.e., only one physical view. This is possible thanks to the explicit diversity regularization that DiCoM enforces on the deep views.

---

### Official Review · Reviewer_UrUc · 2021-10-29

**Correctness:** 3
**Technical Novelty And Significance:** 2
**Empirical Novelty And Significance:** 2
**Recommendation:** 5
**Confidence:** 4

**Main Review:**

1.	The consideration of diversity for semi-supervised learning is interesting, and the reviewer considers that it is helpful for improving performance.
2.	The proposed method combines diversity with consistency based on underlying probabilistic graphical assumptions is a meaningful idea.

weakness
1.	The idea of this paper is direct. This paper still considers traditional semi-supervised learning, and the multi-view refers to the view generated from instance augmentation. Why only measure the diversity of labeled instances and not extend to unlabeled instances? Are there relevant experimental verifications?
2.	What is the advantage of combining diversity with consistency based on underlying probabilistic graphical assumptions? Several multi-view methods have already considered the diversity with consistency simultaneously, but the authors choose to ignore them, e.g., “Exclusivity-Consistency Regularized Multi-view Subspace Clustering”, “End-to-End Adversarial-Attention Network for Multi-Modal Clustering”. More analyses and comparisons are expected, the reviewer considers that these ideas can also be transformed into the semi-supervised framework.
3.	Although this paper considers semi-supervised regression, why can't the semi-supervised classification method extend to the regression problem by replacing the loss function? The SOTA comparison method is in 2018, more comparison methods recently are expected.



**Summary Of The Paper:**

This paper concerns the semi-supervised problem. Different from SOTA deep semi-supervised methods, the proposed DiCom employs a diversity measure on the labeled multi-view data, and combines diversity with consistency based on underlying probabilistic graphical assumptions. Experiments verify the effectiveness.

**Summary Of The Review:**

This paper concerns the semi-supervised problem. Different from SOTA deep semi-supervised methods, the proposed DiCom employs a diversity measure on the labeled multi-view data, and combines diversity with consistency based on underlying probabilistic graphical assumptions. Experiments verify the effectiveness. However, there exist several shortcomings for this paper, so I recommend weak rejection.

---

> ### Author Response · Authors · 2021-11-17
> **Response to Reviewer UrUc (part 1/3)**
>
> We thank the reviewers for the valuable comments. We would like to address each of your concerns as follows.
>
> > 1. The idea of this paper is direct. This paper still considers traditional semi-supervised learning, and the multi-view refers to the view generated from instance augmentation. Why only measure the diversity of labeled instances and not extend to unlabeled instances? Are there relevant experimental verifications?
>
> In the DiCoM loss function, the diversity regularization term is derived from the graphical model in Figure 2c. Since $f_c$ is the only latent variable in the graph, we integrate it out of the joint distribution to obtain a likelihood function that contains only the observed variables $y$ and $\mu$, as shown in equation (7). From there, we apply the ambiguity decomposition [1] to break the likelihood into a supervised term and a diversity term, as seen in equation (9). Thus, the diversity loss can only co-exist with the supervised loss. When the label information is missing, the graphical model in Figure 2c will be incomplete and the diversity loss cannot be derived. As a result, we do not implement diversity regularization on unlabeled data.
>
> Nevertheless, in our experiments, we have performed an ablation study where diversity is extended to unlabeled data. Specifically, we set the weight of the consistency loss term to zero and apply diversity regularization to both labeled and unlabeled training data. This model is called Ablation-4. The results in Table 1 show that DiCoM-N-2 outperforms Ablation-4 in all cases, which suggests that a mere reliance on diversity is insufficient for the learning task. In other words, the consistency regularization is also an important part of our model.
>
> [1] Anders Krogh and Jesper Vedelsby. Neural network ensembles, cross validation and active learning. Advances in Neural Information Processing Systems 7, 7:231, 1995.

---

> ### Author Response · Authors · 2021-11-17
> **Response to Reviewer UrUc (part 2/3)**
>
> > 2. What is the advantage of combining diversity with consistency based on underlying probabilistic graphical assumptions? Several multi-view methods have already considered the diversity with consistency simultaneously, but the authors choose to ignore them, e.g., “Exclusivity-Consistency Regularized Multi-view Subspace Clustering”, “End-to-End Adversarial-Attention Network for Multi-Modal Clustering”. More analyses and comparisons are expected, the reviewer considers that these ideas can also be transformed into the semi-supervised framework.
>
> We thank the reviewer for asking this question. We would like to point out the following insights and advantages of the DiCoM method:
>
> * First, we show that both diversity and consistency can be derived from the same underlying graphical assumption. This means that diversity and consistency are not contradicting with each other.
>
> * Second, we show that diversity should be optimized on labeled data, while consistency should be maximized on unlabeled data. Thus, our method is fundamentally different from other methods such as Semi-supervised NCL [1], that enforce only diversity on *both* labeled and unlabeled data. In our ablation study, we have shown that DiCoM-N performs better than the ablation model that applies only diversity regularization to both labeled and unlabeled data (please refer to Table 1).
>
> * Third, the two forces (diversity and consistency) are not, and should not be treated as equally-weighted components. In fact, our derivations suggest that it depends on the number of views $M$: when $M$ increases, the number of diversity terms grows as $O(M)$, while the number of consistency terms grows as $O(M^2)$. While this insight may not directly help set specific hyperparameter values, we nonetheless observed an increasing trend of $\kappa_{div}/\kappa_{csc}$ ratio as the number of views increases (please refer to Table 2) which qualitatively agrees with our derivations.
>
> * The degree of diversity and consistency can be easily controlled in the DiCoM loss function, which makes it easy to tune for different types of network architectures. We have shown this in our experiments with two variants of DiCoM (multi-network and multi-branch) on two different data types (tabular and visual data).
>
> Regarding the two papers [2,3] suggested by the reviewer: these papers investigate the fundamentally different task and learning setting of unsupervised clustering. While it may be interesting to investigate how to extend these unsupervised clustering methods to semi-supervised regression, this is out of the scope of our work.
>
> [1] Chen, Huanhuan, Bingbing Jiang, and Xin Yao. "Semisupervised negative correlation learning." IEEE transactions on neural networks and learning systems 29.11 (2018): 5366-5379.
>
> [2] Wang, Xiaobo, et al. "Exclusivity-consistency regularized multi-view subspace clustering." Proceedings of the IEEE conference on computer vision and pattern recognition. 2017.
>
> [3] Zhou, Runwu, and Yi-Dong Shen. "End-to-end adversarial-attention network for multi-modal clustering." Proceedings of the IEEE/CVF Conference on Computer Vision and Pattern Recognition. 2020.

---

> ### Author Response · Authors · 2021-11-17
> **Response to Reviewer UrUc (part 3/3)**
>
> > 3. Although this paper considers semi-supervised regression, why can't the semi-supervised classification method extend to the regression problem by replacing the loss function? The SOTA comparison method is in 2018, more comparison methods recently are expected.
>
> We thank the reviewer for the valuable comment.
>
> First, some of the assumptions for semi-supervised classification such as the cluster assumption or the low-density separation assumption do not apply to regression. Nonetheless, we have in fact included comparisons to several adaptations of fairly recent semi-supervised classification methods, namely Mean Teacher and Virtual Adversarial Training by replacing the classification loss with the regression loss. As the results indicate, these methods do not perform well. They are consistently worse than other methods that were designed specifically for regression. This echoes the point that methods designed based on assumptions for semi-supervised classification cannot simply be adapted to regression and be expected to perform well; specialized techniques like ours are required to obtain good performance.
>
> Another issue with state-of-the-art classification methods such as FixMatch [1], UDA [2] or ReMixMatch [3] is that they rely heavily on data augmentation techniques that are specific to visual data only. Nevertheless, we would be glad to include comparisons with other state-of-the-art semi-supervised classification methods if the reviewer provides us with specific suggestions.
>
> We are not aware of any more recent works than SSDKL [4] on deep semi-supervised learning for generic regression tasks. The competing methods that we used in our experiments are considered state-of-the-art in semi-supervised regression. If the reviewer can point us to specific methods that we should include in our comparisons, we would be happy to include them.
>
> [1] Sohn, K., Berthelot, D., Li, C., Zhang, Z., Carlini, N., Cubuk, E., Kurakin, A., Zhang, H., Raffel, C. (2020). FixMatch: Simplifying Semi-Supervised Learning with Consistency and Confidence arXiv https://arxiv.org/abs/2001.07685
>
> [2] Berthelot, D., Carlini, N., Cubuk, E., Kurakin, A., Sohn, K., Zhang, H., Raffel, C. (2019). ReMixMatch: Semi-Supervised Learning with Distribution Alignment and Augmentation Anchoring arXiv https://arxiv.org/abs/1911.09785
>
> [3] Xie, Q., Dai, Z., Hovy, E., Luong, M., Le, Q. (2019). Unsupervised Data Augmentation for Consistency Training arXiv https://arxiv.org/abs/1904.12848
>
> [4] Jean, Neal, Sang Michael Xie, and Stefano Ermon. "Semi-supervised deep kernel learning: Regression with unlabeled data by minimizing predictive variance." arXiv preprint arXiv:1805.10407 (2018).

---

### Official Review · Reviewer_6car · 2021-10-30

**Correctness:** 2
**Technical Novelty And Significance:** 2
**Empirical Novelty And Significance:** 2
**Recommendation:** 3
**Confidence:** 4

**Main Review:**

1. The ablation study missed an interesting question:  would increasing the number of views with either consistency or diversity loss surpass the combined loss with fewer views?

On one hand, increasing the number of views certainly increases the chance of producing more diverse features or predictions, but on the other hand, having more views on the same data also leads to a higher probability of consistent predictions from multiple views. It would be good to see how the number of views plays in the game.

2. The fundamental grounding of multi-view learning is that, the utility of all views is maximised if, given the label, all views are independent of each other.

The paper made an argument in the intro that negative correlations between or among individual views would be helpful, which doesn't make sense. Referred literature in this paper talked about decreasing correlations by including a penalty term, but they didn't mean that we would like to see negative correlations.

3. In ensemble learning, at least in bagging, the improvement over individual learners is often more significant when these learners are more diverse (have larger variance), but again, the baseline model here is a trained base learner. It is highly likely that an ensemble of model A would still be worse than a single model B. There is always a tradeoff between the diversity (variance) of the base learner and its accuracy. I hope the authors could be more precise on their wording in the intro.

4. I am not super positive about the derivations in Section 3. Let me elaborate a bit.

Providing that $$f_m \sim \mathcal{N}(f_c, \sigma_m^2), \forall m=1,...,M$$ with the distribution of f_c unknown.

The joint distribution should be $$p(f_c, f_1, ..., f_M)=p(f_1, ..., f_M|f_c)p(f_c)=p(f_c)\Pi_{m=1}^Mp(f_m|f_c)=p(f_c)\Pi_{m=1}^M\mathcal{N}(f_c, \sigma_n^2)$$, which is different from Eq. 2.

The marginal distribution should be $$p( f_1, ..., f_M)=\int p(f_1, ..., f_M|f_c)p(f_c) df_c=\int p(f_c)\Pi_{m=1}^M\mathcal{N}(f_c, \sigma_m^2)df_c$$, which doesn't give us Eq. 3 since we don't know  the distribution of f_c.

Let's move on to the posterior distribution of the consensus function:

$$p(f_c | f_1, …, f_M) = p(f_c, f_1, …, f_M) / p(f_1, …, f_M)= p(f_c) p(f_1, …, f_M | f_c) / \int p(f_c) p(f_1, …, f_M | f_c) df_c$$

which may not be a gaussian distribution.

A quick fix is to also assume that p(f_c) is a gaussian distribution, although this assumption seems to be very strong.

**Summary Of The Paper:**

the submission proposed a semi-supervised learning framework that leverages the benefits of multi-view learning with neural networks. On labelled data, besides minimising the empirical risk, the objective function encourages diversity by driving the correlation between individual functions with the mean function; on the unlabelled data, the objective forces multiple views to produce consistent features or predictions.

**Summary Of The Review:**

Overall, this submission's empirical contribution is more appreciated than its unclear theoretical justification. However, the arguments made in the intro were too vague and also misleading. Therefore, I do not vote for accepting this paper.

---

> ### Author Response · Authors · 2021-11-17
> **Response to Reviewer 6car (part 1/3)**
>
> We thank the reviewer for carefully reading our paper and for the constructive feedback. Our responses to your comments are given below.
>
> > 1. The ablation study missed an interesting question: would increasing the number of views with either consistency or diversity loss surpass the combined loss with fewer views?
> On one hand, increasing the number of views certainly increases the chance of producing more diverse features or predictions, but on the other hand, having more views on the same data also leads to a higher probability of consistent predictions from multiple views. It would be good to see how the number of views plays in the game.
>
> Thank you for your suggestions. We have conducted an extra experiment to compare the ablation models with $M=4$ views against the DiCoM-N model with $M=2$ views. For Ablation-2, we remove the diversity loss term from the total loss, i.e., diversity is not being enforced at all. For Ablation-3, the consistency term is set to zero. The results are shown in the Table E1 below.
>
> **Table E1. Test RMSE from Extra Ablation Study on UCI Datasets.**
>
> | Dataset    	|   	| DiCoM-N-2         	|   	|   	| Ablation-2 (M=4)  	| % Redc. 	|   	|   	| Ablation-3 (M=4)  	| % Redc.  	|
> |----------	|:-:	|:-----------------:	|:-:	|:-:	|:-----------------:	|:-------:	|:-:	|:-:	|:-----------------:	|:--------:	|
> | skillcraft 	|   	| 0.313 $\pm$ 0.005 	|   	|   	| 0.320 $\pm$ 0.025 	|  -2.077 	|   	|   	| **0.313 $\pm$ 0.012** 	|    0.034 	|
> | parkinsons 	|   	| **2.285 $\pm$ 0.208** 	|   	|   	| 2.415 $\pm$ 0.512 	|  -5.683 	|   	|   	| 2.389 $\pm$ 0.274 	|   -4.555 	|
> | elevators  	|   	| 0.145 $\pm$ 0.025 	|   	|   	| **0.132 $\pm$ 0.018** 	|   8.923 	|   	|   	| 0.134 $\pm$ 0.019 	|    7.679 	|
> | protein    	|   	| 0.646 $\pm$ 0.031 	|   	|   	| **0.631 $\pm$ 0.021** 	|   2.388 	|   	|   	| 0.647 $\pm$ 0.043 	|   -0.078 	|
> | blog       	|   	| 0.930 $\pm$ 0.040 	|   	|   	| **0.914 $\pm$ 0.020** 	|   1.738 	|   	|   	| 0.945 $\pm$ 0.040 	|   -1.584 	|
> | ctslice    	|   	| **5.575 $\pm$ 0.606** 	|   	|   	| 6.417 $\pm$ 0.575 	| -15.098 	|   	|   	| 6.620 $\pm$ 0.826 	|  -18.733 	|
> | buzz       	|   	| 0.715 $\pm$ 0.136 	|   	|   	| **0.673 $\pm$ 0.040** 	|   5.952 	|   	|   	| 0.709 $\pm$ 0.049 	|    0.870 	|
> | electric   	|   	| 0.114 $\pm$ 0.025   	|   	|   	| 0.136 $\pm$ 0.068   	| -18.854 	|   	|   	| **0.088 $\pm$ 0.006**   	|   22.667 	|
> | Average    	|   	|                   	|   	|   	|                   	|  -2.839 	|   	|   	|                   	|    0.787 	|
>
> On average, Ablation-2 with more views is still worse (-2.84%) than the full DiCoM-N model with fewer views, while Ablation-3 is slightly better (0.79%). Thus, adding more views alone does not provide a sufficient level of diversity compared to explicitly enforcing it through the diversity loss. It is interesting to see that the Ablation-3 model performs better than the Ablation-2 model. The same trend can be observed from Table 1 in the main paper. This suggests that promoting diversity contributes more toward learning than enforcing consistency.

---

> ### Author Response · Authors · 2021-11-17
> **Response to Reviewer 6car (part 2/3)**
>
> > 2. The fundamental grounding of multi-view learning is that, the utility of all views is maximised if, given the label, all views are independent of each other.
> The paper made an argument in the intro that negative correlations between or among individual views would be helpful, which doesn't make sense. Referred literature in this paper talked about decreasing correlations by including a penalty term, but they didn't mean that we would like to see negative correlations.
>
> We would like to thank the reviewer for the comment.
> The reviewer has described the *conditional independence assumption*, which states that the views should be conditionally independent, given the true label; this assumption was originally proposed in the Co-training paper [1]. However, it is not a necessary assumption in multi-view learning, because other multi-view techniques such as Canonical Correlation Analysis [2] or Multiple Kernel Learning [3] do not rely on this assumption. Even for co-training style algorithms, this conditional independence assumption was found to be too strong to be used in practice. Thus, several alternative assumptions have been proposed to relax it, namely the weak dependence assumption [4], the expansion assumption [5] and the large diversity assumption [6].
>
> Having said that, a more generic condition for the success of multi-view learning is the complementary principle [7], which states that each view should contain some knowledge that other views do not have, so that multiple views can be employed to comprehensively and accurately describe the data. Our DiCoM method is firmly grounded in this principle through the use of the diversity loss, which is based on negative correlation. Here, the term negative correlation refers to the correlation among the *errors* of the individual learners, not among the learners’ outputs themselves. When the ensemble output is the average of the learners’ outputs, having learners with negatively correlated errors is better than having them uncorrelated [8]. Thus, our work is well-motivated by the literature in both multi-view learning and ensemble learning.
>
> To avoid confusion, we would like to correct one statement in the Related Work section. In the original submission, we wrote “When this penalty term is maximized, the individual learners become negatively correlated.”. In Revision-1, to clarify this point, that sentence has been changed to “When this penalty term is maximized, the errors of individual learners become negatively correlated.”.
>
> [1] Blum, Avrim, and Tom Mitchell. "Combining labeled and unlabeled data with co-training." Proceedings of the eleventh annual conference on Computational learning theory. 1998.
>
> [2] Hotelling, Harold. "Relations Between Two Sets of Variates." Biometrika 28.3/4 (1936): 321-377.
>
> [3] Bach, Francis R., Gert RG Lanckriet, and Michael I. Jordan. "Multiple kernel learning, conic duality, and the SMO algorithm." Proceedings of the twenty-first international conference on Machine learning. 2004.
>
> [4] Abney, Steven. "Bootstrapping." Proceedings of the 40th Annual Meeting on Association for Computational Linguistics. Association for Computational Linguistics, 2002.
>
> [5] Balcan, Maria-Florina, Avrim Blum, and Ke Yang. "Co-training and expansion: Towards bridging theory and practice." Advances in neural information processing systems. 2004.
>
> [6] Wang, Wei, and Zhi-Hua Zhou. "Analyzing co-training style algorithms." Machine Learning: ECML 2007. Springer Berlin Heidelberg, 2007. 454-465.
>
> [7] Xu, Chang, Dacheng Tao, and Chao Xu. "A survey on multi-view learning." arXiv preprint arXiv:1304.5634 (2013).
>
> [8] Clemen, Robert T., and Robert L. Winkler. "Limits for the precision and value of information from dependent sources." Operations Research 33.2 (1985): 427-442.

---

> ### Author Response · Authors · 2021-11-17
> **Response to Reviewer 6car (part 3/3)**
>
> > 3. In ensemble learning, at least in bagging, the improvement over individual learners is often more significant when these learners are more diverse (have larger variance), but again, the baseline model here is a trained base learner. It is highly likely that an ensemble of model A would still be worse than a single model B. There is always a tradeoff between the diversity (variance) of the base learner and its accuracy. I hope the authors could be more precise on their wording in the intro.
>
> We thank the reviewer for this helpful comment. In the Introduction section of the original manuscript, we said “Ensemble methods have been shown to be better than individual models if the ensemble is both accurate and diversified (Dietterich, 2000).”. In Revision-1, we have changed the sentence to “A necessary and sufficient condition for an ensemble of learners to be more accurate than any of its individual members is if the base learners are accurate and diverse (Dietterich, 2000).” to be more precise.
>
>
> > 4. I am not super positive about the derivations in Section 3. Let me elaborate a bit.
> Providing that...
>
> We thank the reviewer for asking this question and we apologize for any confusion. We would like to clarify that our graphical models are undirected graphical models; we have also further clarified this in Revision-1. In an undirected graphical model, the edges do not represent causal relationships between nodes, but rather the correlation or affinity between them. Thus, the joint distribution of an undirected graphical model is proportional to the product of the potential functions of the maximal cliques of the graph. If the graph has no clique (which is the case in our DiCoM model), the joint distribution can be written as the normalized product of pairwise potential functions, i.e., edge potentials. In our case, we have $M$ edges connecting $f_c$ with each of the deep view $f_m$, each edge potential is an isotropic un-normalized Gaussian potential on the difference between $f_c$ and $f_m$. Thus, we can write the joint distribution as a normalized product of these Gaussian potentials.
>
> Taken in this proper context of undirected graphical models, our mathematical derivations in the Proposed Method section are correct. To enhance clarity, we have made changes to clarify these details in equation (2) and listed all the potential functions in Figure 2.

---

### Official Review · Reviewer_BTsB · 2021-11-02

**Correctness:** 4
**Technical Novelty And Significance:** 2
**Empirical Novelty And Significance:** 2
**Recommendation:** 5
**Confidence:** 3

**Main Review:**

1. I want to know how many branches is good and why don't you use multiple heads? e.g., GAT uses multi-head.
2. In multiview learning, I want to know how to model diversity? The consistency is easy to model, but the diversity is not well addressed. Even CCA can be regarded as a view-consistency modeling.
3. I want to know why you design the model? for what applications?

**Summary Of The Paper:**

A view-consistent loss is designed, which is based on PGM. The PGM is quite simple to understand.

**Summary Of The Review:**

The view-consistency modeling does not give a supervise or somehow it is not new enough.
The diversity loss is not well addressed.

---

> ### Author Response · Authors · 2021-11-17
> **Response to Reviewer BTsB**
>
> We thank the reviewer for the valuable comments. Our answers to your questions are as follows.
>
> > 1. I want to know how many branches is good and why don't you use multiple heads? e.g., GAT uses multi-head.
>
> The optimal number of branches depends on various factors, including the dimensionality and variance of the label and the complexity of the learning task. In our experiments, we have tried $M = 2, 4, 8$ branches, which is smaller than the typical number of base learners in ensemble learning. We have performed an analysis to study the effect of the number of views in the Experiment section (please refer to Figure 4). Our results show that the reductions in test errors seem to be plateauing at $M=8$ views. We believe that the number of views in DiCoM does not need to be large thanks to the effectiveness of its diversity regularization. Thus, we suggest that a small $M$ (less than 10 views) can be sufficient to achieve good performance with DiCoM.
>
> Regarding Graph Attention Networks (GAT), GAT and DiCoM are fundamentally different in that GAT applies to data with a graph structure, while DiCoM is a generic method that does not have any restriction on data structure. That being said, the way GAT employs multi-head attention is related to the way DiCoM enables diversity. Specifically, GAT uses multiple independent attention mechanisms to transform the input features and concatenates/averages these transformations to obtain the output features. This multi-head design increases the information richness of the feature representation, as long as the transformations of multiple heads are independent. In DiCoM, we use a multi-branch architecture (see Figure 1b), where each branch is expected to be diverse from each other. This diversity is enforced by optimizing the diversity term in the DiCoM loss function. In fact, the multi-head design of GAT can be considered as a special case of multi-branch designs, because if we branch out at the second-to-last layer, we will effectively have a multi-head architecture. In the crowd counting experiment, our DiCoM-B-4 model branches out at the penultimate layer. Thus, we have actually demonstrated DiCoM on a multi-head architecture.
>
>
> > 2. In multiview learning, I want to know how to model diversity? The consistency is easy to model, but the diversity is not well addressed. Even CCA can be regarded as a view-consistency modeling.
>
> In conventional multi-view learning, the views are usually constructed from physically distinct sets of features, e.g., images or words describing the same object can be used as two views. Naturally, each view carries some information that the other views do not contain. Another example is multiple kernel learning, where multiple views are constructed from different kernels corresponding to different notions of similarity. In general, conventional multi-view learning achieves diversity through a bottom-up approach, starting from the input or the feature level.
>
> The diversity regularization in our DiCoM does not follow this approach. Unlike the conventional methods, we explicitly design a loss function to induce diversity among the different views. In other words, we enforce diversity via a top-down approach: from the loss function down to hidden features using backpropagation. We found this approach flexible and suitable for various implementations of neural networks because both consistency and diversity can be controlled within the loss function.
>
>
> > 3. I want to know why you design the model? for what applications?
>
> We thank the reviewer for asking this question. Our DiCoM method is designed for the generic semi-supervised regression task. There is no specific application that was targeted. As far as we know, there is not much work in the area of generic deep semi-supervised regression. In our experiments, we have demonstrated the efficacy of DiCoM on two data modalities: tabular data and visual data.

---

### Author Response · Authors · 2021-11-17
**Revision-1 Updates**

We thank the reviewers for providing us with valuable feedback. We have attempted to address all questions and concerns that were raised by the reviewers, and have revised our manuscript accordingly. The changes in this revision, namely Revision-1, are as follows:
* In the Introduction section, thanks to reviewer 6car, one statement has been corrected.
* In the Related Work section, one sentence has been edited based on reviewer 6car’s suggestion.
* In the Proposed Method section, we have addressed reviewer C8H3’s concern about equation (1).
* In the Proposed Method section, equation (2) has been modified according to the comments from reviewers 6car and C8H3.
* In the Proposed Method section, Figure 2 has been changed based on reviewer C8H3’s suggestions.
* In the Experiment section, to address questions from reviewer C8H3, we have added a new sentence in the experiment setup and a new paragraph to explain the data augmentation process.
* In the Experiment section, for the ablation study experiment, two sentences have been edited for correctness.

---

> ### Author Response · Authors · 2021-11-30
> **Revision-1 Follow-up**
>
> Again, we thank the reviewers for providing us with valuable comments, which helped improve the quality of our manuscript.
>
> As the final stage of discussion is coming to an end (Nov 29), we would be glad to receive from the reviewers any additional feedback on our paper, or any further follow-up discussion to our first-round responses.
>
> We look forward to hearing from you soon.
>
> Thank you.

---

### Decision · Program_Chairs · 2022-01-20

**Decision:**

Reject

**Comment:**

The work proposed multi-view learning framework that combines diversity and consistency objectives for semi-supervised learning. While reviewers appreciated that simplicity of the proposed method, they raised concerns on the limited contribution on top of the original Bayesian Co-Training work. Although authors provided detailed rebuttals that addressed some of the reviewers' concerns, and one reviewer did raise their score, the other reviewers' scores remained unchanged. Given the work is closely based off the BCT work, I would like to see more detailed analyses on the importance of the changes brought in this work, such as changing the base learners and introduction of diversity objectives as pointed out by the authors.